# POPGym: Benchmarking Partially Observable Reinforcement Learning

**Steven Morad, Ryan Kortvelesy, Matteo Bettini, Stephan Liwicki, Amanda Prorok**

## Abstract

Real world applications of Reinforcement Learning (RL) are often partially observable, thus requiring memory. Despite this, partial observability is still largely ignored by contemporary RL benchmarks and libraries. We introduce Partially Observable Process Gym (POPGym), a two-part library containing (1) a diverse collection of 15 partially observable environments, each with multiple difficulties and (2) implementations of 13 memory model baselines – the most in a single RL library. Existing partially observable benchmarks tend to fixate on 3D visual navigation, which is computationally expensive and only one type of POMDP. In contrast, POPGym environments are diverse, produce smaller observations, use less memory, and often converge within two hours of training on a consumer-grade GPU. We implement our high-level memory API and memory baselines on top of the popular RLlib framework, providing plug-and-play compatibility with various training algorithms, exploration strategies, and distributed training paradigms. Using POPGym, we execute the largest comparison across RL memory models to date. POPGym is available at https://github.com/proroklab/popgym.

## 1 Introduction

Datasets like MNIST (Lecun et al., 1998) have driven advances in Machine Learning (ML) as much as new architectural designs (Levine et al., 2020). Comprehensive datasets are paramount in assessing the progress of learning algorithms and highlighting shortcomings of current methodologies. This is evident in the context of RL, where the absence of fast and comprehensive benchmarks resulted in a reproducability crisis (Henderson et al., 2018). Large collections of diverse environments, like the Arcade Learning Environment, OpenAI Gym, ProcGen, and DMLab provide a reliable measure of progress in deep RL. These fundamental benchmarks play a role in RL equivalent to that of MNIST in supervised learning (SL).

The vast majority of today's RL benchmarks are designed around Markov Decision Processes (MDPs). In MDPs, agents observe a *Markov state*, which contains all necessary information to solve the task at hand. When the observations are Markov states, the Markov property is satisfied, and traditional RL approaches guarantee convergence to an optimal policy (Sutton & Barto, 2018, Chapter 3). But in many RL applications, observations are ambiguous, incomplete, or noisy – any of which makes the MDP *partially observable* (POMDP) (Kaelbling et al., 1998), breaking the Markov property and all convergence guarantees. Furthermore, Ghosh et al. (2021) find that policies trained under the ideal MDP framework cannot generalize to real-world conditions when deployed, with epistemic uncertainty turning real-world MDPs into POMDPs. By introducing *memory* (referred to as sequence to sequence models in SL), we can summarize the observations[1] therefore restoring policy convergence guarantees for POMDPs (Sutton & Barto, 2018, Chapter 17.3).

Despite the importance of memory in RL, most of today's comprehensive benchmarks are fully observable or near-fully observable. Existing partially observable benchmarks are often navigation-based – representing only spatial POMDPs, and ignoring applications like policymaking, disease diagnosis, teaching, and ecology (Cassandra, 1998). The state of memory-based models in RL libraries is even more dire – we are not aware of any RL libraries that implement more than three

---

[1]Strictly speaking, the agent actions are also required to guarantee convergence. We consider the previous action as part of the current observation.

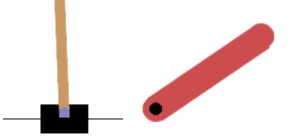
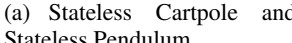
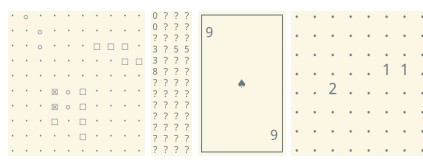
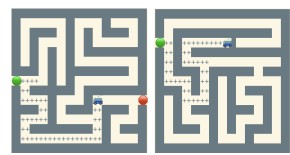

(a) Stateless Cartpole and Stateless Pendulum

(b) Battleship, Concentration, Higher Lower and Mine Sweeper

(c) Labyrinth Escape and Explore

Figure 1: Renders from select POPGym environments.

or four distinct memory baselines. In nearly all cases, these memory models are limited to frame stacking and LSTM.

To date, there are no popular RL libraries that provide a diverse selection of memory models. Of the few existing POMDP benchmarks, even fewer are comprehensive and diverse. As a consequence, there are no large-scale studies comparing memory models in RL. We propose to fill these three gaps with our proposed POPGym.

## 1.1 CONTRIBUTIONS

POPGym is a collection of 15 partially observable gym environments (Figure 1) and 13 memory baselines. All environments come with at least three difficulty settings and randomly generate levels to prevent overfitting. The POPGym environments use low-dimensional observations, making them fast and memory efficient. Many of our baseline models converge in under two hours of training on a single consumer-grade GPU ( Table 1, Figure 2). The POPGym memory baselines utilize a simple API built on top of the popular RLlib library (Liang et al., 2018), seamlessly integrating memory models with an assortment of RL algorithms, sampling, exploration strategies, logging frameworks, and distributed training methodologies. Utilizing POPGym and its memory baselines, we execute a large-scale evaluation, analyzing the capabilities of memory models on a wide range of tasks. To summarize, we contribute:

1. A comprehensive collection of diverse POMDP tasks.
2. The largest collection of memory baseline implementations in an RL library.
3. A large-scale, principled comparison across memory models.

## 2 RELATED WORK

There are many existing RL benchmarks, which we categorize as fully (or near-fully) observable and partially observable. In near-fully observable environments, large portions of the the Markov state are visible in an observation, though some information may be missing. We limit our literature review to *comprehensive* benchmarks (those that contain a wide set of tasks), as environment diversity is essential for the accurate evaluation of RL agents (Cobbe et al., 2020).

### 2.1 FULLY AND NEAR-FULLY OBSERVABLE BENCHMARKS

The Arcade Learning Environment (ALE) (Bellemare et al., 2013) wraps Atari 2600 ROMs in a Python interface. Most of the Atari games, such as Space Invaders or Missile Command are fully observable (Cobbe et al., 2020). Some games like asteroids require velocity observations, but models can recover full observability by stacking four consecutive observations (Mnih et al., 2015), an approach that does not scale for longer timespans. Even seemingly partially-observable multi-room games like Montezuma's Revenge are made near-fully observable by displaying the player's score and inventory (Burda et al., 2022).

OpenAI Gym (Brockman et al., 2016) came after ALE, implementing classic fully observable RL benchmarks like CartPole and MountainCar. Their Gym API found use in many other environments, including our proposed benchmark.

Cobbe et al. (2020) find that randomly generated environments are critical to training general agents, showing policies will overfit to specific levels otherwise. They propose ProcGen: 16 procedurally generated environments with pixel-space observations. Most environments are fully or near-fully observable, although a few environments provide a partially observable mode, effectively turning them into 2D area coverage (navigation) tasks. ProcGen motivates POPGym's use of random level generation.

## 2.2 Partially Observable Benchmarks

When enumerating partially observable benchmarks, we find many are based on 3D first-person navigation. DeepMind Lab (Beattie et al., 2016) (DMLab) is a 3D first-person view navigation simulator based on the Quake 3 physics engine. It implements various tasks such as collecting fruits, maze exploration, and laser tag. VizDoom (Kempka et al., 2016) is another 3D navigation simulator based on the PC game Doom. It gives the agent weapons and adds computer-controlled characters that can shoot at the player. Miniworld (Chevalier-Boisvert, 2018) is a third 3D first-person view navigation simulator that is easier to install than DMLab or VizDoom. MiniGrid (Chevalier-Boisvert et al., 2018) and GridVerse (Baisero & Katt, 2021) are 2D navigation simulators with a first-person view. Unlike the previously mentioned 3D simulators, agents converge on gridworld environments much faster due to the smaller observation space. This makes it a popular benchmark for memory models.

There are few POMDP libraries that provide tasks beyond navigation. Behaviour suite (BSuite) evaluates agents on a variety of axes, one of which is memory (Osband et al., 2020), but they only provide two POMDPs. Similar to our benchmark, (Zheng & Tellex, 2020) provide classic POMDPs with low-dimensional observation spaces. But their tasks are solvable without neural networks and are not difficult enough for modern deep RL. Ni et al. (2022) provide 21 environments, most of which are a special case of POMDP known as *latent MDPs* (Kwon et al., 2021), where a specific MDP is chosen from a set of possible MDPs at the beginning of an episode. (Morad et al., 2022) provides three POMDPs, which is insufficient for a benchmark.

We briefly mention the Starcraft (Samvelyan et al., 2019) and VMAS (Bettini et al., 2022) benchmarks because multi-agent environments are intrinsically partially observable, but we focus specifically on single-agent POMDPs.

## 2.3 Shortcomings of Current Benchmarks

Popular fully observable benchmarks use pixel-based observation spaces, adding a layer of complexity that takes an order of magnitude longer to train when compared against state-based observation counterparts (Seita, 2020). In fully observable environments, visually pleasing results are worth a few extra hours training. This dogma persists into partial observability, where environments often take 10x longer to converge than their fully observable counterparts. Popular benchmarks using 3D graphics take hundreds of billions of timesteps (Parisotto et al., 2020) and multiple weeks (Morad et al., 2021) on a GPU to train a single policy to convergence. Until sample efficiency in partially observable RL improves, we must forgo pixel-based observations or continue to struggle with reproducibility.

Many partially observable tasks with pixel-based observation spaces are based on some form of navigation (Ramani, 2019). Although navigation can be a partially observable task, wall following behavior in perfect mazes guarantees complete area coverage without the need for memory. When mazes are imperfect (i.e. contain cycles), deterministic wall following can get stuck in infinite loops. However, RL policies often have some amount of stochasticity that can break out of these loops. Kadian et al. (2020) and Morad et al. (2021) inadvertently show that memory-free navigation agents learn wall following strategies[2] that are surprisingly effective in imperfect real-world mazes. We confirm these findings with our experiments, showing that memory-free agents are competitive with memory-endowed agents in certain navigation benchmarks.

All other (imperfect) mazes can be fully explored by storing no more than two past locations (observations) in memory (Blum & Kozen, 1978). Navigation-based tasks like area coverage, moving to a coordinate, or searching for items can be reduced to the maze exploration task. We do not claim

---

[2]https://en.wikipedia.org/wiki/Maze-solving_algorithm#Wall_follower

that navigation tasks are easy, but rather that it is important to have a variety of tasks to ensure we evaluate all facets of memory, such as *memory capacity*, that navigation tasks might miss.

### 2.4 EXISTING MEMORY BASELINES

The state of memory models in RL is even more bleak than the benchmarks. Most libraries provide frame stacking and a single type of RNN. OpenAI Baselines (Dhariwal et al., 2017), Stable-Baselines3 (Raffin et al., 2021), and CleanRL (Huang et al., 2021) provide implementations of PPO with frame stacking and an LSTM. Ray RLlib (Liang et al., 2018) currently implements frame stacking, LSTM, and a transformer for some algorithms. Ni et al. (2022) implement LSTM, GRUs, and two model-based memory models. Yang & Nguyen (2021) provides recurrent versions of the DDPG, TD3, and SAC RL algorithms, which utilize GRUs and LSTM. Zheng & Tellex (2020) implement multiple classical POMDP solvers, but these do not use neural networks, preventing their application to more complex tasks. There is currently no go-to library for users who want to compare or apply non-standard memory models.

### 2.5 A BRIEF SUMMARY ON MEMORY

When designing a library of memory models, it is important to select competitive models. Ni et al. (2022) show that sequence to sequence models from SL are competitive or better than RL-specific memory methods while being more straightforward to implement, so we focus specifically on sequence to sequence models (called memory throughout the paper). Although a strict categorization of memory is elusive, most methods are based on RNNs, attention, or convolution.

RNNs (Elman, 1990) take an input and hidden state, feed them through a network, and produce a corresponding output and hidden state. RNNs depend on the previous state and must be executed sequentially, resulting in slow training but fast inference when compared with other methods.

Attention-based methods (Vaswani et al., 2017) have supplanted RNNs in many applications of SL, but traditional transformers have quadratically-scaling memory requirements, preventing them from running on long episodes in RL. Recent linear attention formulations (Schlag et al., 2021; Katharopoulos et al., 2020) claim to produce transformer-level performance in linear time and space, potentially enabling widespread use of attention in RL.

Like attention, convolutional methods are computationally efficient (Bai et al., 2018), lending themselves well to RL. They are less common than recurrent or attention-based methods in SL, and there is little literature on their use in RL.

## 3 POPGYM ENVIRONMENTS

All of our environments bound the cumulative episodic reward in $[-1, 1]$. In some cases (e.g. repeating previous observations) an optimal policy would receive a cumulative reward of one in expectation. In other environments (e.g. playing battleship with randomly placed ships), an optimal policy has an expected episodic cumulative reward of less than one.

We tag our proposed environments as *diagnostic*, *control*, *noisy*, *game*, and *navigation*. Each tag is designed to represent a different class of POMDP, and each environment has at least three distinct difficulty settings, creating the most diverse POMDP benchmark thus far. Our proposed environments are all *overcomplete* POMDPs, meaning our environments have more unique latent Markov states than unique observations (Sharan et al., 2017; Jin et al., 2020).

**Diagnostic** environments probe model capabilities with respect to the duration of memories, forgetting, and compression and recall. They are designed to quickly summarize the strengths and weaknesses of a specific model.

**Control** environments are control RL environments made partially observable by removing part of the observation. Solving these tasks only requires short-term memory.

**Noisy** environments require the memory model to estimate the true underlying state by computing an expectation over many observations. These are especially useful for real-world robotics tasks.

**Game** environments provide a more natural and thorough evaluation of memory through card and board games. They stress test memory capacity, duration, and higher-level reasoning.

**Navigation** environments are common in other benchmarks, and we include a few to ensure our benchmark is comprehensive. More than anything, our navigation environments examine how memory fares over very long sequences.

## 3.1 ENVIRONMENT DESCRIPTIONS

1. **Repeat First (Diagnostic):** At the first timestep, the agent receives one of four values and a remember indicator. Then it randomly receives one of the four values at each successive timestep without the remember indicator. The agent receives a reward for outputting (remembering) the first value.

2. **Repeat Previous (Diagnostic):** Like Repeat First, observations contain one of four values. The agent is rewarded for outputting the observation from some constant $k$ timesteps ago, i.e. observation $o_{t-k}$ at time $t$.

3. **Autoencode (Diagnostic):** During the WATCH phase, a deck of cards is shuffled and played in sequence to the agent with the watch indicator set. The watch indicator is unset at the last card in the sequence, where the agent must then output the sequence of cards in order. This tests whether the agent can encode a series of observations into a latent state, then decode the latent state one observation at a time.

4. **Stateless Cartpole (Control):** The cartpole environment from Barto et al. (1983), but with the angular and linear positions removed from the observation. The agent must integrate to compute positions from velocity.

5. **Stateless Pendulum (Control):** The swing-up pendulum (Doya, 1995), with the angular position information removed.

6. **Noisy Stateless Cartpole (Control, Noisy):** Stateless Cartpole (Env. 4) with Gaussian noise.

7. **Noisy Stateless Pendulum (Control, Noisy):** Stateless Pendulum (Env. 5) with Gaussian noise.

8. **Multiarmed Bandit (Noisy, Diagnostic):** The multiarmed bandit problem (Slivkins & others, 2019; Lattimore & Szepesvári, 2020) posed as an episodic task. Every episode, bandits are randomly initialized. Over the episode, the player must trade off exploration and exploitation, remembering which bandits pay best. Each bandit has some probability of paying out a positive reward, otherwise paying out a negative reward. Note that unlike the traditional multiarmed bandit task where the bandits are fixed once initialized, these bandits reset every episode, forcing the agent to learn a policy that can adapt between episodes.

9. **Higher Lower (Game, Noisy):** Based on the card game higher-lower, the agent must guess if the next card is of higher or lower rank than the previous card. The next card is then flipped face-up and becomes the previous card. Using memory, the agent can utilize card counting strategies to predict the expected value of the next card, improving the return.

10. **Count Recall (Game, Diagnostic, Noisy):** Each turn, the agent receives a next value and query value. The agent must answer the query with the number of occurrences of a specific value. In other words, the agent must store running counts of each unique observed value, and report a specific count back, based on the query value. This tests whether the agent can learn a compressed structured memory representation, such that it can continuously update portions of memory over a long sequence.

11. **Concentration (Game):** A deck of cards is shuffled and spread out face down. The player flips two cards at a time face up, receiving a reward if the flipped cards match. The agent must remember the value and position of previously flipped cards to improve the rate of successful matches.

12. **Battleship (Game):** A partially observable version of Battleship, where the agent has no access to the board and must derive its own internal representation. Observations contain either HIT or MISS and the position of the last salvo fired. The player receives a positive reward for striking a ship, zero reward for hitting water, and negative reward for firing on a specific tile more than once.

Table 1: Frames per second (FPS) of our environments, computed on the Google Colab free tier and a Macbook Air (2020) laptop.

| Environment | Colab FPS | Laptop FPS | Environment | Colab FPS | Laptop FPS |
|---|---|---|---|---|---|
| Repeat First | 23,895 | 155,201 | Multiarmed Bandit | 48,751 | 469,325 |
| Repeat Previous | 50,349 | 136,392 | Battleship | 117,158 | 235,402 |
| Autoencode | 121,756 | 251,997 | Concentration | 47,515 | 157,217 |
| Stateless Cartpole | 73,622 | 218,446 | Higher Lower | 24,312 | 76,903 |
| Stateless Pendulum | 8,168 | 26,358 | Count Recall | 16,799 | 53,779 |
| Noisy Stateless Cartpole | 6,269 | 66,891 | Minesweeper | 8,434 | 32,003 |
| Noisy Stateless Pendulum | 6,808 | 20,090 | Labyrinth Escape | 1,399 | 41,122 |
| | | | Labyrinth Explore | 1,374 | 30,611 |

13. **Mine Sweeper (Game):** The computer game Mine Sweeper, but like our Battleship implementation, the agent does not have access to the board. Each observation contains the position and number of adjacent mines to the last square "clicked" by the agent. Clicking on a mined square will end the game and produce a negative reward. The agent must remember where it has already searched and must integrate information from nearby tiles to narrow down the location of mines. Once the agent has selected all non-mined squares, the game ends.

14. **Labyrinth Explore (Navigation):** The player is placed in a discrete, 2D procedurally-generated maze, receiving a reward for each previously unreached tile it reaches. The player can only observe adjacent tiles. The agent also receives a small negative reward at each timestep, encouraging the agent to reach all squares quickly and end the episode.

15. **Labyrinth Escape (Navigation):** The player must escape the procedurally-generated maze, using the same observation space as Labyrinth Explore. This is a sparse reward setting, where the player receives a positive reward only after solving the maze.

## 4 POPGYM BASELINES

Our memory model API relies on an abstract memory model class, only requiring users to implement `memory_forward` and `initial_state` methods. Our memory API builds on top of RLlib, exposing various algorithms, exploration methods, logging, distributed training, and more.

We collect well-known memory models from SL domains and wrap them in this API, enabling their use on RL tasks. We rewrite models where the existing implementation is slow, unreadable, not amenable to our API, or not written in Pytorch. Some of these sequence models have yet to be applied in the context of reinforcement learning.

1. **MLP:** An MLP that cannot remember anything. This serves to form a lower bound for memory performance, as well and ensuring memory models are actively using memory, rather than just leveraging their higher parameter counts.

2. **Positional MLP (PosMLP):** An MLP that can access the current episodic timestep. The current timestep is fed into the positional encoding from Vaswani et al. (2017), which is summed with the incoming features. PosMLP enables agents to learn time-dependent policies (those which evolve over the course of an episode) without explicitly using memory.

3. **Elman Networks:** The original RNN, from Elman (1990). Elman networks sum the recurrent state and input, passing the resulting vector through a linear layer and activation function to produce the next hidden state. Elman networks are not used much in SL nowadays due to vanishing and exploding gradients.

4. **Long Short-Term Memory (LSTM):** Hochreiter & Schmidhuber (1997) designed LSTM to address the vanishing and exploding gradient problems present in earlier RNNs like the Elman Network. LSTM utilizes a constant error carousel to handle longer dependencies and gating to ensure recurrent state stability during training. It has two recurrent states termed the hidden and cell states.

5. **Gated Recurrent Unit (GRU):** The GRU is a simplification of LSTM, using only a single recurrent state. The GRU appears to have similar performance to LSTM in many applications while using fewer parameters (Chung et al., 2014).

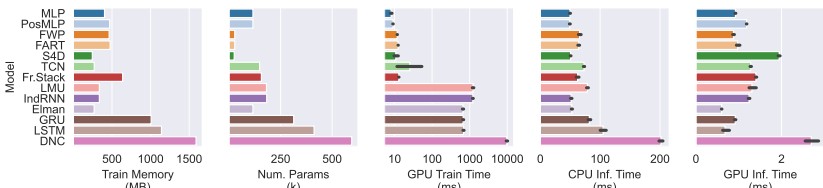

Figure 2: Performance characteristics for POPGym memory baselines on random inputs. We use a recurrent state size of 256, a batch size of 64, and a episode length of 1024. We compute CPU statistics on a 3GHz Xeon Gold and GPU statistics on a 2080Ti, reporting the mean and 95% confidence interval over 10 trials. Train times correspond to a full batch while inference times are per-element (i.e. the latency to compute a single action). Note that GPU Train Time has logarithmic scale.

6. **Independently Recurrent Networks (IndRNN):** Stacking LSTM and GRU cells tends to provide few benefits when compared with traditional deep neural networks. IndRNNs update the recurrent state using elementwise connections rather than a dense layer, enabling much deeper RNNs and handling longer dependencies than the LSTM and GRU (Li et al., 2018). In our experiments, we utilize a 2-layer IndRNN.

7. **Differentiable Neural Computers (DNC):** Graves et al. (2016) introduce a new type of recurrent model using external memory. The DNC utilizes an RNN as a memory controller, reading and writing to external storage in a differentiable manner.

8. **Fast Autoregressive Transformers (FART):** Unlike the traditional attention matrix whose size scales with the number of inputs, FART computes a fixed-size attention matrix at each timestep, taking the cumulative elementwise sum over successive timesteps (Katharopoulos et al., 2020). FART maintains two recurrent states, one for the running attention matrix and one for a normalization term, which helps mitigate large values and exploding gradients as the attention increases grows over time. The original paper omits a positional encoding, but we find it necessary for our benchmark.

9. **Fast Weight Programmers (FWP):** The theory behind FART and FWP is different, but the implementation is relatively similar. FWP also maintains a running cumulative sum. Unlike FART, FWP normalizes the key and query vectors to sum to one, requiring only a single recurrent state and keeping the attention matrix of reasonable scale (Schlag et al., 2021). Unlike the original paper, we add a positional encoding to FWP.

10. **Frame Stacking (Fr.Stack):** Mnih et al. (2015) implemented frame stacking to solve Atari games. Frame stacking is the concatenation of $k$ observations along the feature dimension. Frame stacking is not strictly convolutional, but is implemented similarly to other convolutional methods. Frame stacking is known to work very well in RL, but the number of parameters scales with the receptive field, preventing it from learning long-term dependencies.

11. **Temporal Convolutional Networks (TCN):** TCNs slide 1D convolutional filters over the temporal dimension. On long sequences, they are faster and use less memory than RNNs. TCNs avoid the vanishing gradient problem present in RNNs because the gradient feeds through each sequence element individually, rather than propagating through the entire sequence (Bai et al., 2018).

12. **Legendre Memory Units (LMU):** LMUs are a mixture of convolution and RNNs. They apply Legendre polynomials across a sliding temporal window, feeding the results into an RNN hidden state (Voelker et al., 2019). LMUs can handle temporal dependencies spanning up to 100K timesteps.

13. **Diagonal State Space Models (S4D):** S4D treats memory as a controls problem. It learns a linear time invariant (LTI) state space model for the recurrent state. S4D applies a Vandermonde matrix to the sequence of inputs, which we can represent using either convolution or a recurrence. Computing the result convolutionally makes it very fast. In SL, S4D was able to solve the challenging 16,000 timestep Path-X task, demonstrating significant capacity for long-term dependencies (Gu et al., 2022).

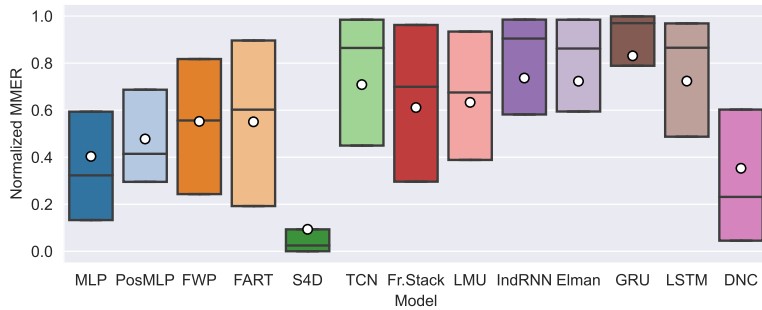

| Model | MMER | MMER w/o Nav |
|---|---|---|
| MLP | 0.067 | -0.010 |
| PosMLP | 0.064 | 0.053 |
| FWP | 0.112 | 0.200 |
| FART | 0.138 | 0.202 |
| S4D | -0.180 | -0.119 |
| TCN | 0.233 | 0.219 |
| Fr.Stack | 0.190 | 0.177 |
| LMU | 0.229 | 0.246 |
| IndRNN | 0.259 | 0.302 |
| Elman | 0.249 | 0.224 |
| **GRU** | **0.349** | **0.326** |
| LSTM | 0.255 | 0.294 |
| DNC | 0.065 | 0.016 |

Figure 3: (Left) A summary comparison of baselines aggregated over all environments. We normalize the MMER such that 0 denotes the worst trial and 1 denotes the best trial for a specific environment. We report the interquartile range (box), median (horizontal line), and mean (dot) normalized MMER over all trials. (Right) Single value scores for each model, produced by meaning the MMER over all POPGym environments. We also provide scores with navigation (Labyrinth) environments excluded; the reasoning is provided in the discussion section.

## 5 EXPERIMENTS

Our memory framework hooks into RLlib, providing integration with IMPALA, DQN, and countless other algorithms. Due to computational constraints, we only execute our study on Proximal Policy Optimization (PPO) (Schulman et al., 2017). We tend to use conservative hyperparameters to aid in reproducability – this entails large batch sizes, low learning rates, and many minibatch passes over every epoch. We run three trials of each model over three difficulties for each environment, resulting in over 1700 trials. We utilize the *max-mean episodic reward* (MMER) in many plots. We compute MMER by finding the mean episodic reward for each epoch, then taking the maximum over all epochs, resulting in a single MMER value for each trial. We present the full experimental parameters in Appendix A and detailed results for each environment and model in Appendix B. We provide a summary over models and tasks in Figure 3. Figure 2 reports model throughput and Table 1 provides environment throughput.

## 6 DISCUSSION

In the following paragraphs, we pose some questions and findings made from the results of our large-scale study.

**Supervised learning is a bad proxy for RL.** Supervised learning experiments show that IndRNN, LMU, FART, S4D, DNC, and TCN surpass LSTM and GRUs by a wide margin (Li et al., 2018; Voelker et al., 2019; Katharopoulos et al., 2020; Gu et al., 2022; Graves et al., 2016; Bai et al., 2018). S4D is unstable and often crashed due to exploding weights, suggesting it is not suitable for RL out of the box and that further tuning may be required. Although linear attention methods like FWP and FART show significant improvements across a plethora of supervised learning tasks, they were some of the worst contenders in RL. Classical RNNs outperformed modern memory methods, even though RNNs have been thoroughly supplanted in SL (Figure 3). The underlying cause of the disconnect between RL and SL performance is unclear and warrants further investigation.

**Use GRUs for performance and Elman nets for efficiency.** Within traditional RNNs, there seems little reason to use LSTM, as GRUs are more efficient and perform better. Elman networks are largely ignored in modern SL and RL due to vanishing or exploding gradients, but these issues did not impact our training. We find Elman networks perform on-par with LSTM while exhibiting some of the best parameter and memory efficiency out of any model (Figure 2). Future work could investigate why Elman networks work so well in RL given their limitations, and distill these properties into memory models suited specifically for RL.

**Are maze navigation tasks sufficient for benchmarking memory?** Existing POMDP benchmarks focus primarily navigation tasks. In our experiments, we show that the MLP received the

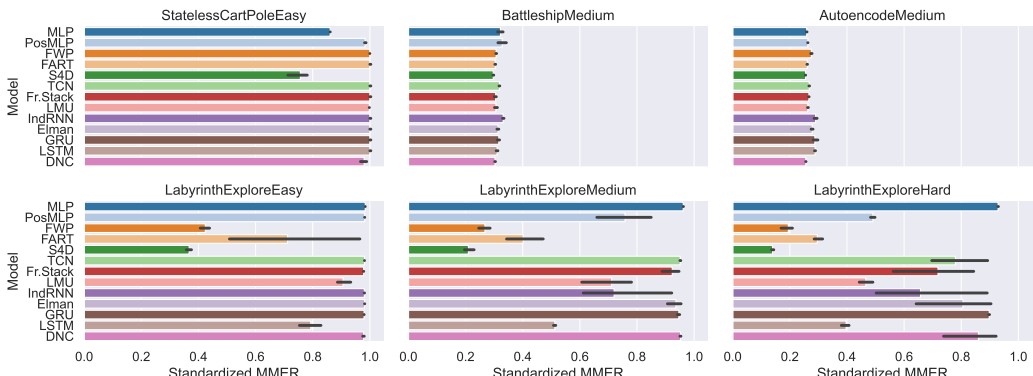

Figure 4: Selected results used in the discussion section. We standardize the MMER from $[-1, 1]$ to $[0, 1]$ for readability. The colored bars denote the mean and the black lines denote the 95% bootstrapped confidence interval. Full results across all environments are in Appendix B

highest score on almost all navigation tasks, beating all memory models (Figure 4). This is in line with our hypothesis from subsection 2.3, and raises doubts concerning previous models evaluated solely on navigation tasks. Does a novel memory method outperform baselines because of a better memory architectures, or just because it has more trainable parameters? Future work can bypass this scrutiny by including a diverse set of tasks beyond navigation, and by modifying simple navigation tasks to better leverage memory (e.g. positive reward for correctly answering "how many rooms are there in the house?").

**Positional MLPs are an important baseline.** Masked control tasks turn MDPs into POMDPs by hiding the velocity or position portions of classic control problems, and are probably the second most popular type of POMDP in literature after navigation. The positional MLP performed notably better than the MLP, nearly solving the Stateless Cartpole masked control task on easy (Figure 4). This is entirely unexpected, as providing the current timestep to an MLP is insufficient to compute the position and underlying Markov state. Outside of masked control, the positional MLP regularly outperformed the MLP (Figure 3). Stateless policies that evolve over time could be an interesting topic for future work, and should be a standard baseline in future memory comparisons.

**Is PPO enough?** Memory models do not noticeably outperform the MLP in many game environments, such as Autoencode or Battleship, indicating that the memory is minimally effective in these tasks (Figure 4). All thirteen models converge to the nearly same reward, suggesting this could be due to issues with PPO rather than the memory models themselves. Future work could focus on designing new algorithms to solve these tasks. In parallel, research institutions with ample compute could ablate POPGym across other algorithms, such as Recurrent Replay Distributed DQN (Kapturowski et al., 2019).

## 7  CONCLUSION

We presented the POPGym benchmark, a collection of POMDPs and memory baselines designed to standardize RL in partially observable environments. We discovered a notable disconnect between memory performance in supervised and reinforcement learning, with older RNNs surpassing linear transformers and modern memory models. According to our experiments, the GRU is the best general-purpose memory model, with Elman networks providing the best tradeoff between performance and efficiency. We revealed shortcomings in prior benchmarks focused on control and navigation POMDPs, emphasizing the importance of numerous and diverse POMDPs for evaluating memory. There is still a great deal of work to be done towards solving POMDPs, and we hope POPGym provides some measure of progress along the way.

## 8  ACKNOWLEDGEMENTS

Steven Morad and Stephan Liwicki gratefully acknowledge the support of Toshiba Europe Ltd. Ryan Kortvelesy was supported by Nokia Bell Labs through their donation for the Centre of Mobile, Wearable Systems and Augmented Intelligence to the University of Cambridge.

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

Table 2: PPO hyperparameters used in all of our experiments.

| HParam | Value |
|---|---|
| Decay factor $\gamma$ | 0.99 |
| Value fn. loss coef. | 1.0 |
| Entropy loss coef. | 0.0 |
| Learning rate | 5e-5 |
| Num. SGD iters | 30 |
| Batch size | 65536 |
| Minibatch size | 8192 |
| GAE $\lambda$ | 1.0 |
| KL target | 0.01 |
| KL coefficient | 0.2 |
| PPO clipping | 0.3 |
| Value clipping | 0.3 |
| BPTT Truncation Length | $\infty$ |
| Maximum Episode Length | 1024 |

## A  EXPERIMENTAL PARAMETERS

Given the number of environments and models, it is not computationally feasible to optimize hyper-parameters in a structured way. Through trial and error, we evaluated all models on the Repeat First and Repeat Previous environments and found suitable values that worked across all models. We then picked a more conservative estimate (larger batch size, lower learning rate) to promote monotonic improvement and prevent catastrophic forgetting, at the expense of some sample efficiency.

There is no clear axis for a truly fair comparison between memory models – model throughput and parameter count vary drastically throughout the memory models. We decide to limit the amount of storage (i.e. recurrent state size) of each memory model to 256 dimensions, which is greater than the common values of 64 and 128 in literature (Ni et al., 2022). This results in lower parameter counts for models that produce the recurrent state using a tensor product (e.g. FWP, FART, S4D). We could make an exception for these models, allowing them to produce recurrent states of size $256^2 = 63356$ dimensions instead of $16^2 = 256$ dimensions to bring up the parameter count, but we believe this is unfair to recurrent models. In this case, recurrent models would need to forget and compress information over longer episodes, while tensor product models could store every single observation in memory without any compression or forgetting. Storing everything is unlikely to scale to real-world applications where episodes could span hours, days, or run indefinitely.

### A.1  GENERAL MODEL HYPERPARAMETERS

For all our memory experiments, we use the same outer model, just swapping out the inner memory model. The outer model first projects observations from the environment to 128 zero-mean variance-one dimensions. Here, the positional encoding is applied if the memory model requests it. The projection goes through a single linear layer and LeakyReLU activation of size 128, then feeds into the memory model. Output from the memory model is projected to 128 dimensions, then split into the actor and critic head. The actor and critic heads are two layer MLPs of width 128 with LeakyReLU activation.

### A.2  MODEL-SPECIFIC HYPERPARAMETERS

The Elman, GRU, LSTM, and LMU RNNs use a single cell. We use a 2-cell IndRNN as they claim to utilize deeper networks. The FART and FWP models use a single attention block. We use the attention-only formulations of FWP, rather than the combined attention and RNN variant. We utilize a temporal window of four for frame stacking and TCN. LMU utilizes a $\theta$ window size of 64 timesteps.

## B   FULL EXPERIMENTAL RESULTS

We report our results in three forms:

1. Bar charts denoting the standardized MMER split by environment (Figure 5-Figure 9)
2. Line plots showing cumulative maximum episodic reward for each training epoch, split by environment (Figure 10-Figure 14)
3. Tables reporting the mean and standard deviation of the MMER, split by model and environment (Table 3)

All models and environments are from commit e397e5e except for the DiffNC experiments, which are from commit 33b0995. All experiments sample 15M timesteps from each environment, except for the DiffNC experiments which sample 10M timesteps.

We run each trial 3 times, aggregating results using the mean over trials. All raw data is available at https://wandb.ai/prorok-lab/popgym-public. The bar plots represent the mean and 95% bootstrap confidence interval. For the bar charts, we standardize the reward between 0 and 1. In the line plots, the solid region refers to the mean and the shaded region to the 95% bootstrap confidence interval. The table reports the MMER mean and standard deviation across trials.

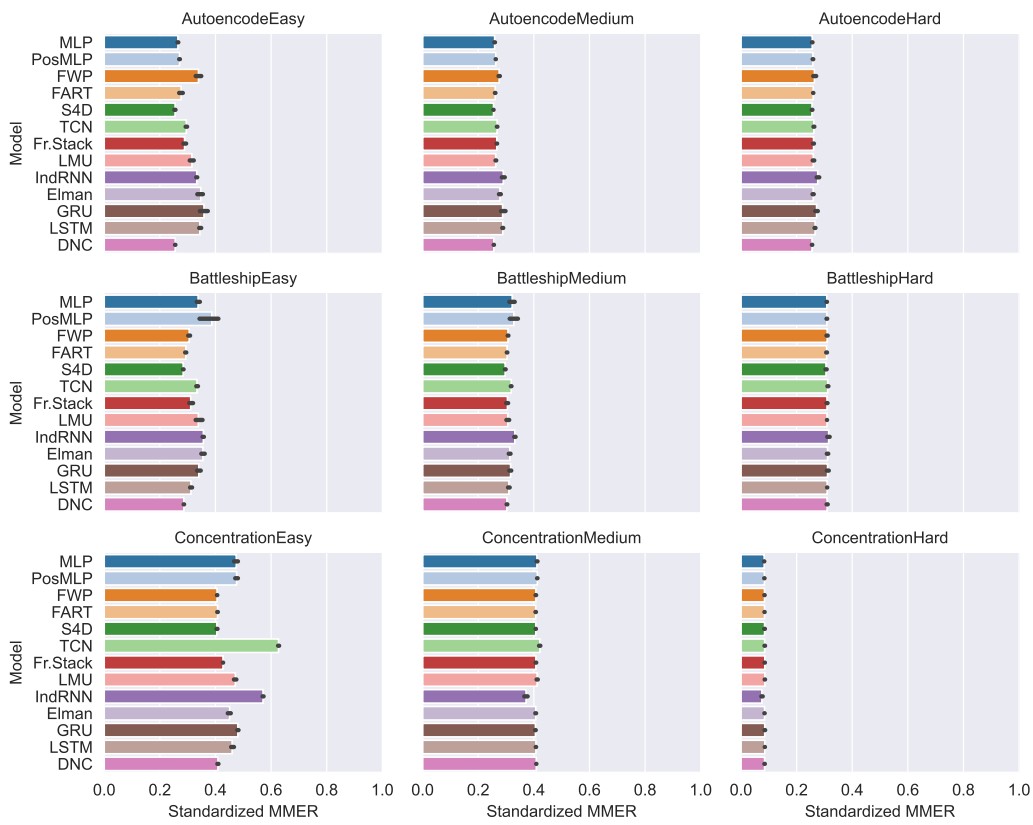

Figure 5: POPGym baselines.

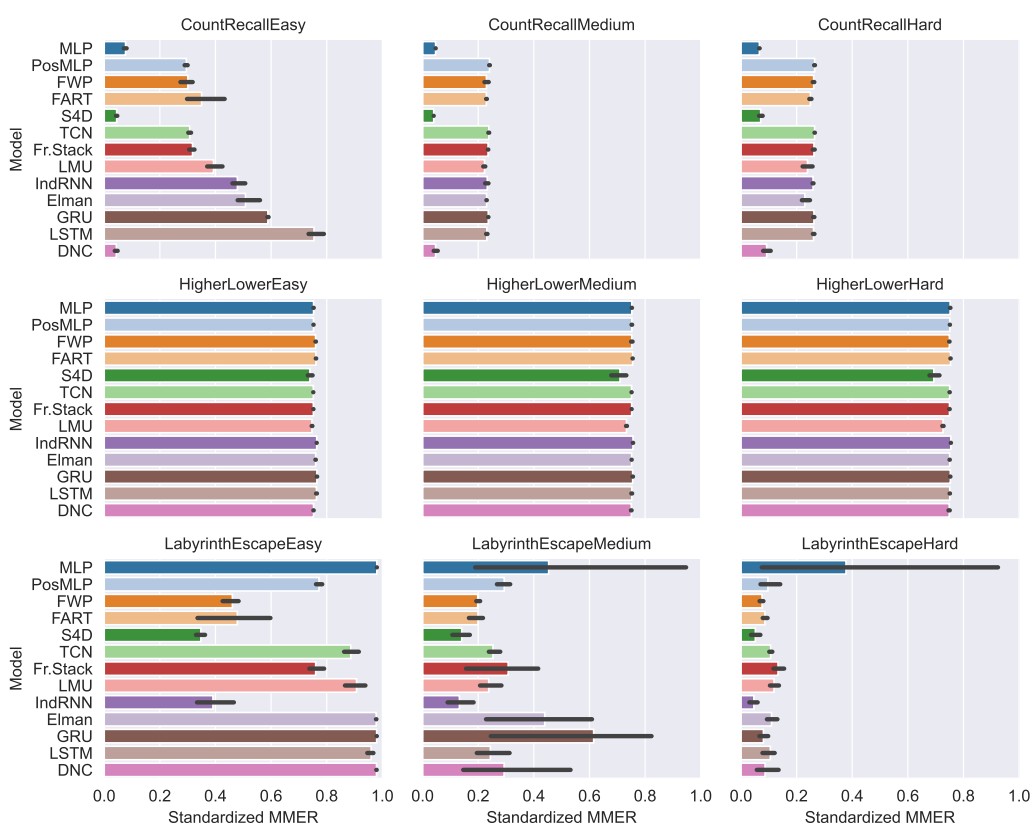

Figure 6: POPGym baselines (continued)

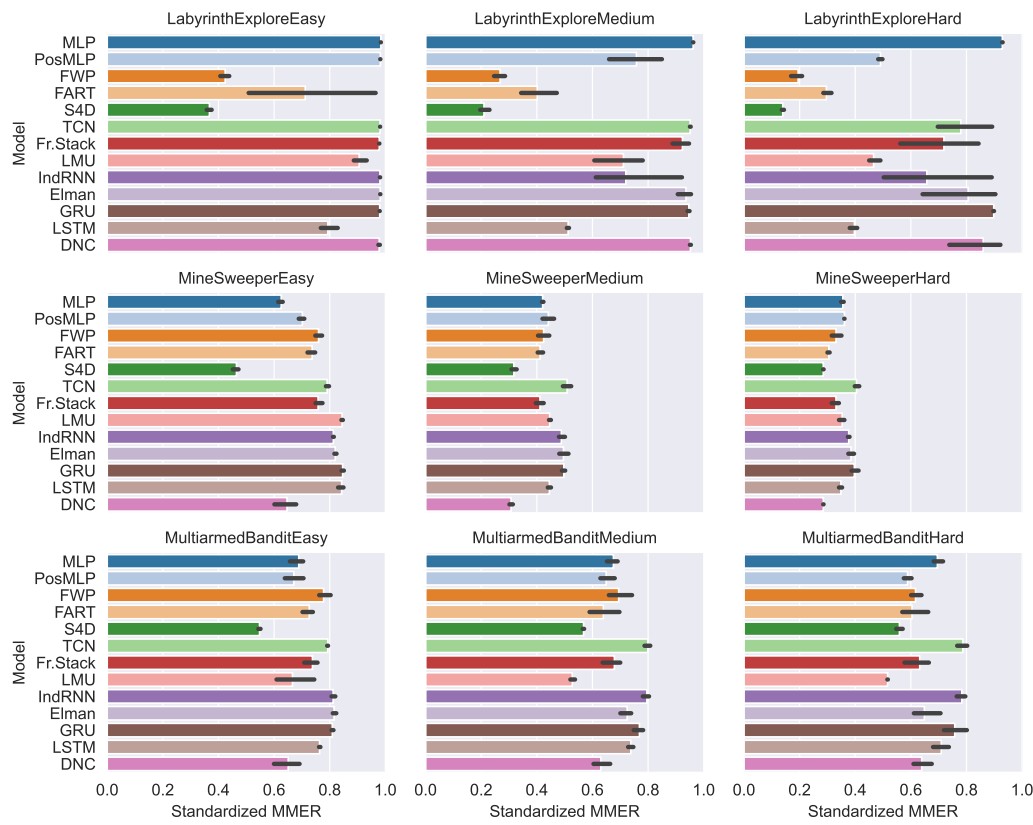

Figure 7: POPGym baselines (continued)

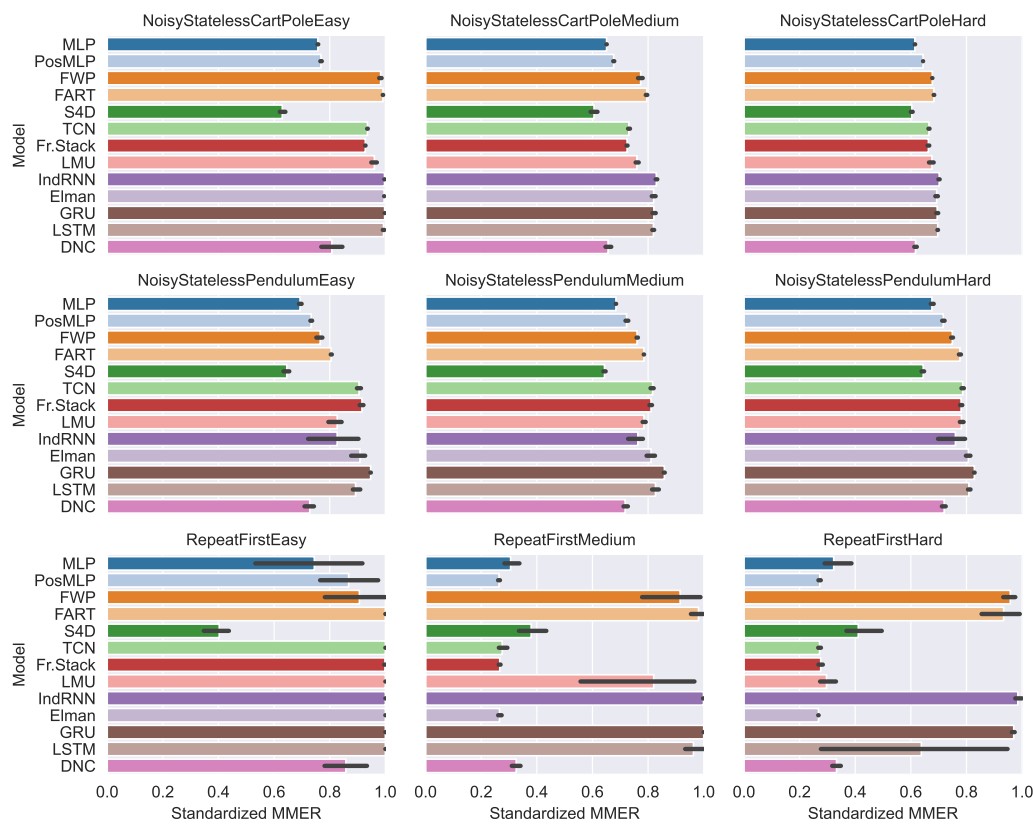

Figure 8: POPGym baselines (continued)

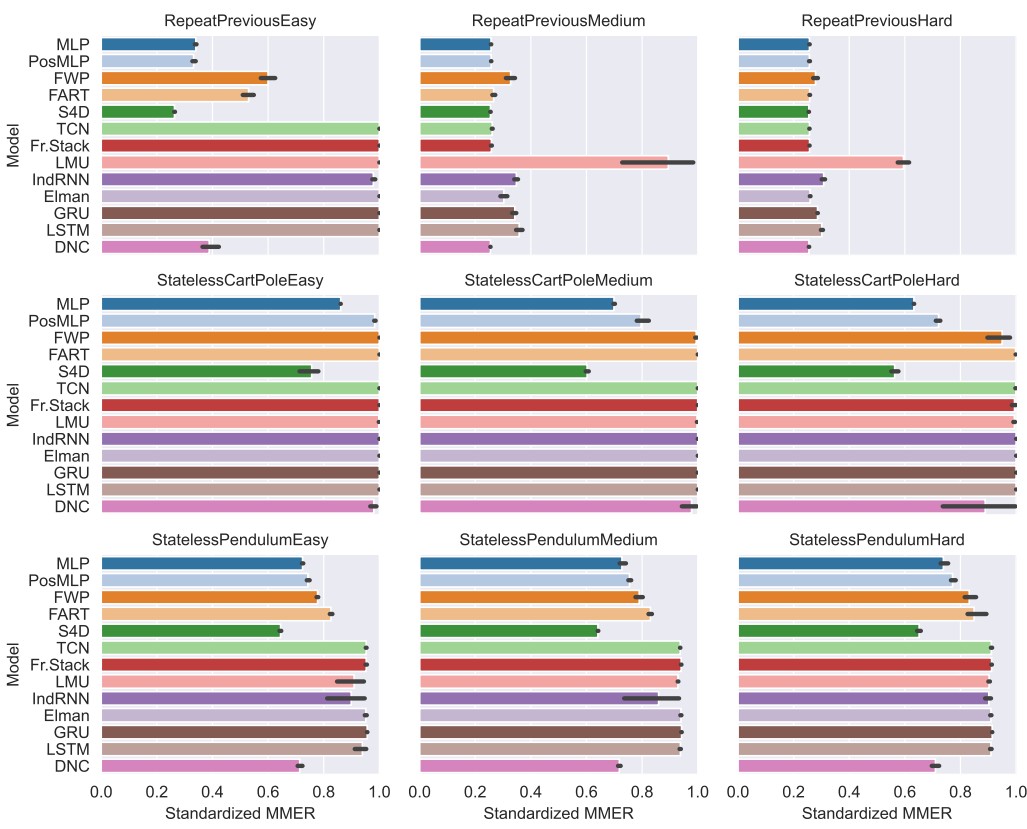

Figure 9: POPGym baselines (continued)

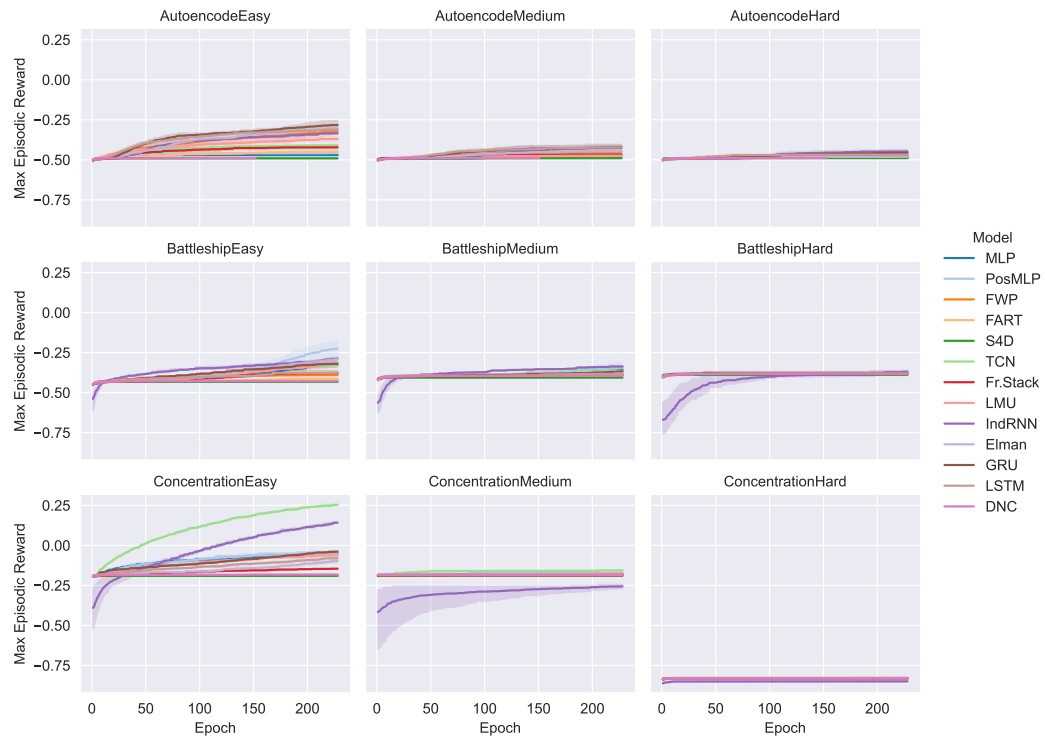

Figure 10: POPGym baselines (continued)

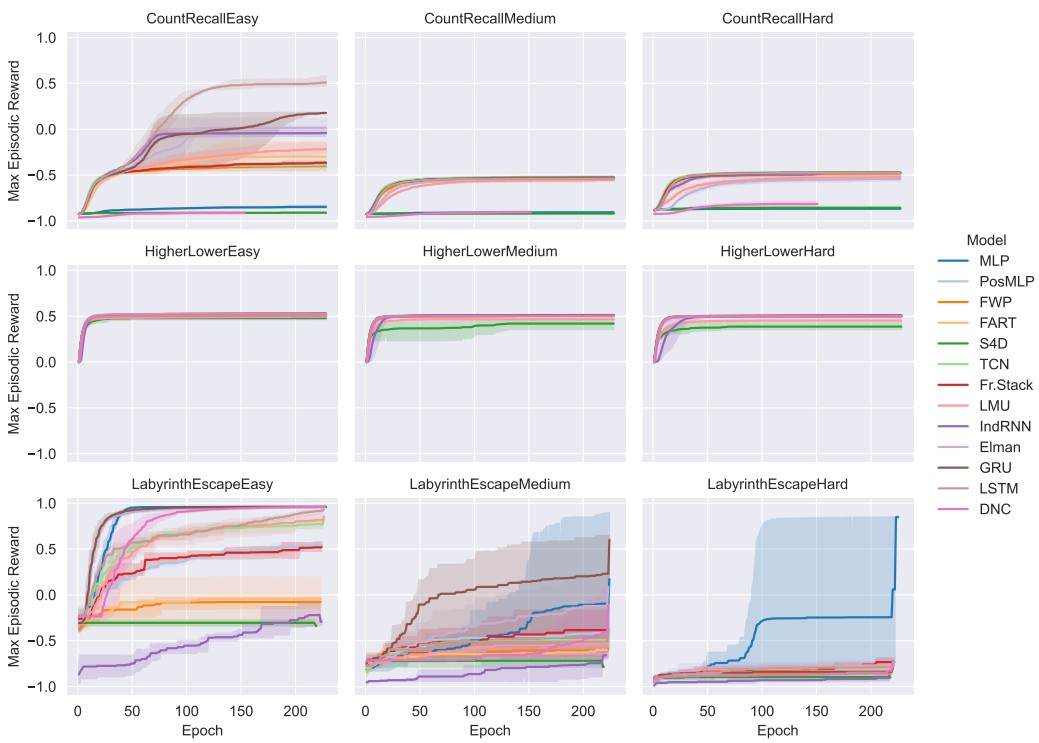

Figure 11: POPGym baselines (continued)

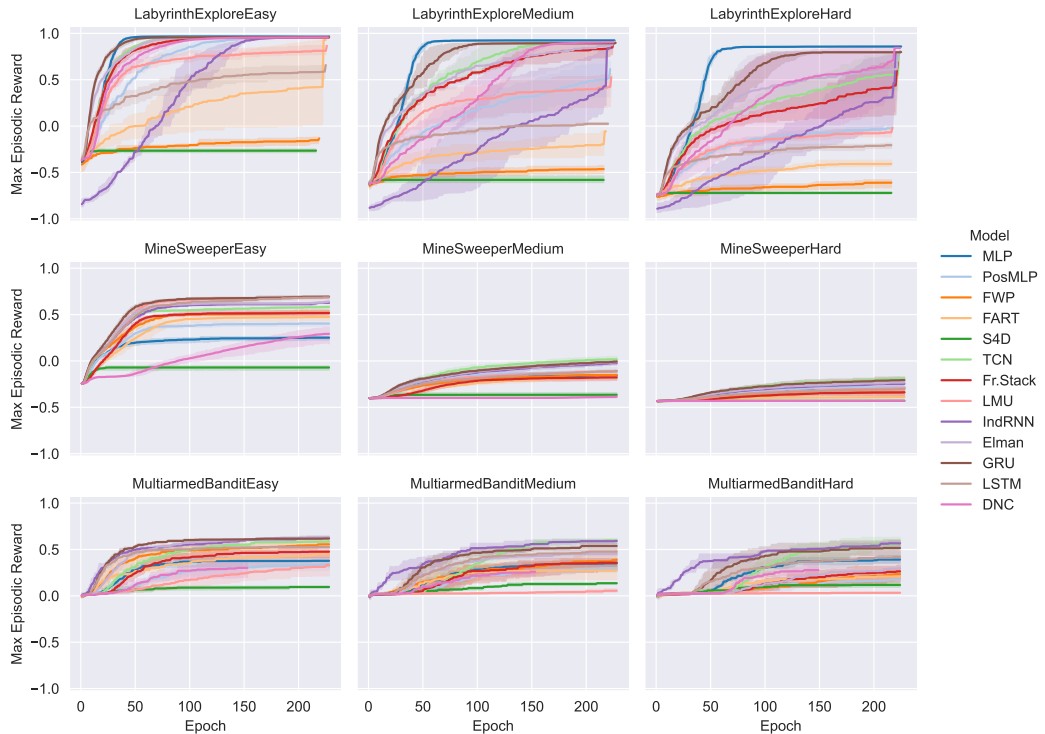

Figure 12: POPGym baselines (continued)

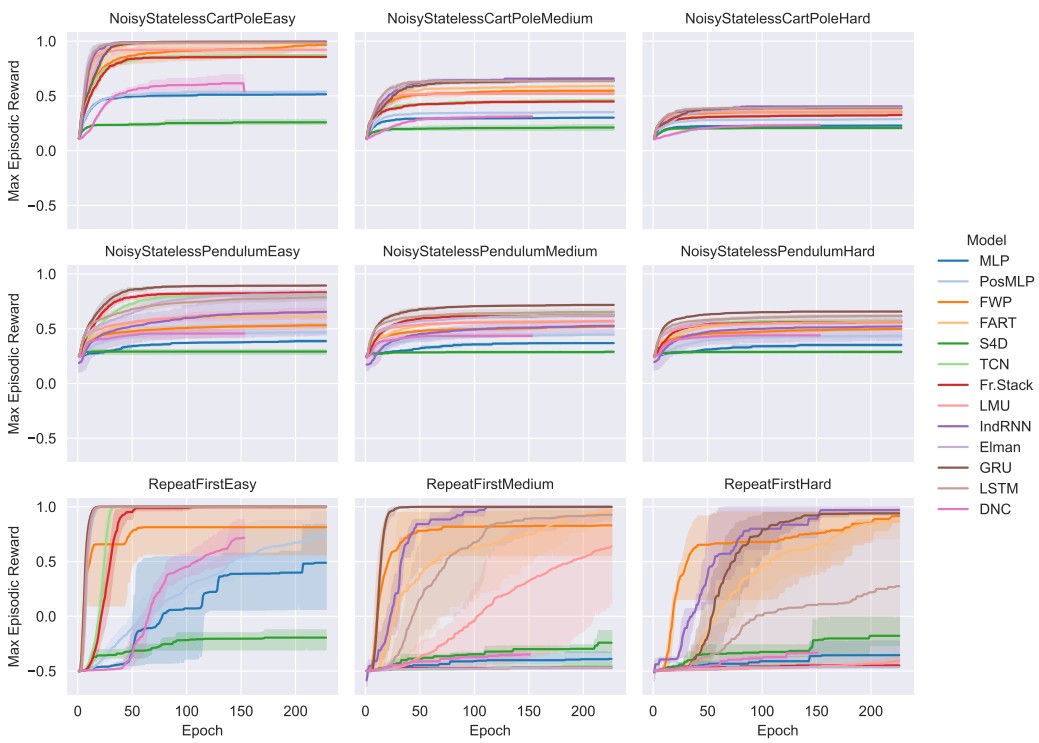

Figure 13: POPGym baselines (continued)

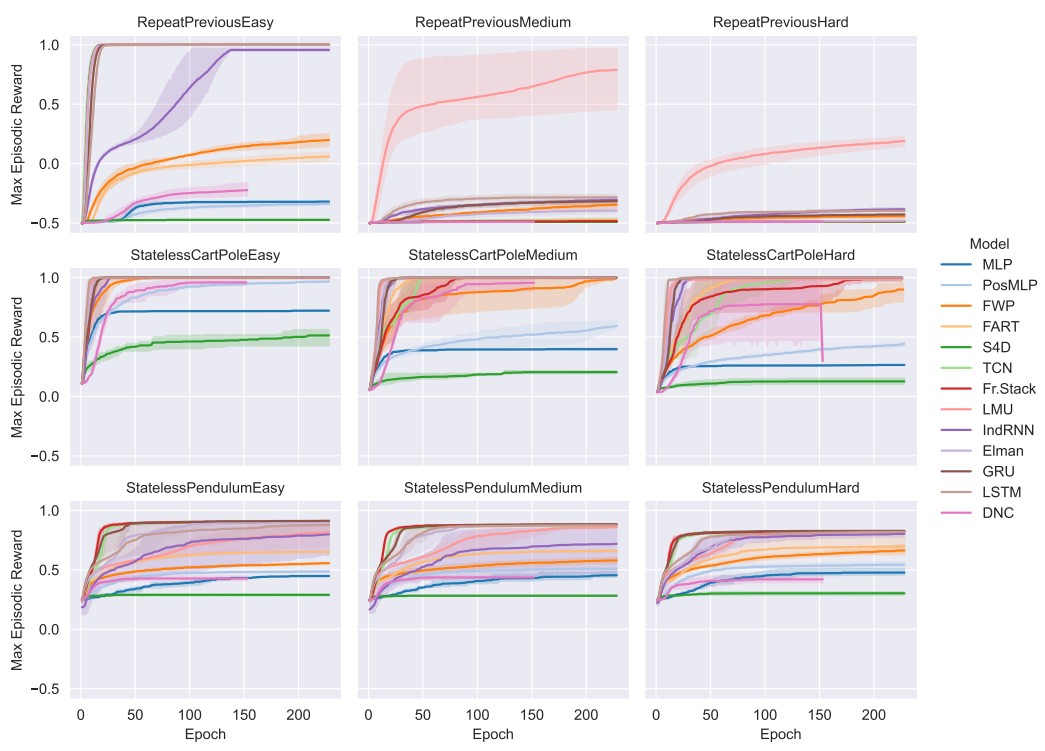

Figure 14: POPGym baselines (continued)

Table 3: Full table of results, denoting the MMER mean and standard deviation by environment and model.

| Env. | Model | MMER | |
|---|---|---|---|
| | | $\mu$ | $\sigma$ |
| AutoencodeEasy | DNC | -0.489 | 0.002 |
| | Elman | -0.306 | 0.022 |
| | FART | -0.447 | 0.014 |
| | FWP | -0.322 | 0.020 |
| | Fr.Stack | -0.422 | 0.010 |
| | **GRU** | **-0.283** | **0.029** |
| | IndRNN | -0.334 | 0.004 |
| | LMU | -0.370 | 0.015 |
| | LSTM | -0.312 | 0.008 |
| | MLP | -0.470 | 0.004 |
| | PosMLP | -0.458 | 0.003 |
| | S4D | -0.490 | 0.005 |
| | TCN | -0.410 | 0.006 |
| AutoencodeMedium | DNC | -0.488 | 0.002 |
| | Elman | -0.443 | 0.007 |
| | FART | -0.478 | 0.002 |
| | FWP | -0.449 | 0.005 |
| | Fr.Stack | -0.466 | 0.002 |
| | GRU | -0.425 | 0.018 |
| | **IndRNN** | **-0.420** | **0.011** |
| | LMU | -0.474 | 0.003 |
| | **LSTM** | **-0.423** | **0.004** |
| | MLP | -0.482 | 0.002 |
| | PosMLP | -0.474 | 0.001 |
| | S4D | -0.490 | 0.001 |
| | TCN | -0.464 | 0.002 |
| AutoencodeHard | DNC | -0.489 | 0.002 |
| | Elman | -0.481 | 0.005 |
| | FART | -0.481 | 0.001 |
| | FWP | -0.472 | 0.011 |
| | Fr.Stack | -0.478 | 0.004 |
| | GRU | -0.456 | 0.009 |
| | **IndRNN** | **-0.448** | **0.010** |
| | LMU | -0.480 | 0.006 |
| | LSTM | -0.467 | 0.004 |
| | MLP | -0.488 | 0.002 |
| | PosMLP | -0.483 | 0.003 |
| | S4D | -0.489 | 0.003 |
| | TCN | -0.476 | 0.004 |
| BattleshipEasy | DNC | -0.427 | 0.002 |
| | Elman | -0.290 | 0.013 |
| | FART | -0.413 | 0.005 |
| | FWP | -0.389 | 0.007 |
| | Fr.Stack | -0.378 | 0.015 |
| | GRU | -0.320 | 0.013 |
| | IndRNN | -0.287 | 0.005 |
| | LMU | -0.323 | 0.027 |
| | LSTM | -0.376 | 0.007 |
| | MLP | -0.325 | 0.012 |
| | **PosMLP** | **-0.226** | **0.077** |
| | S4D | -0.432 | 0.002 |

Continued on next page

|  | | $\mu$ | $\sigma$ |
|---|---|---|---|
| Env. | Model | | |
| | TCN | -0.333 | 0.007 |
| BattleshipMedium | DNC | -0.394 | 0.003 |
| | Elman | -0.373 | 0.007 |
| | FART | -0.392 | 0.003 |
| | FWP | -0.386 | 0.003 |
| | Fr.Stack | -0.390 | 0.008 |
| | GRU | -0.367 | 0.008 |
| | **IndRNN** | **-0.337** | **0.005** |
| | LMU | -0.387 | 0.010 |
| | LSTM | -0.379 | 0.006 |
| | MLP | -0.356 | 0.020 |
| | PosMLP | -0.344 | 0.030 |
| | S4D | -0.406 | 0.003 |
| | TCN | -0.363 | 0.003 |
| BattleshipHard | DNC | -0.380 | 0.004 |
| | Elman | -0.377 | 0.005 |
| | FART | -0.384 | 0.003 |
| | FWP | -0.380 | 0.005 |
| | Fr.Stack | -0.381 | 0.004 |
| | GRU | -0.377 | 0.008 |
| | **IndRNN** | **-0.369** | **0.009** |
| | LMU | -0.381 | 0.000 |
| | LSTM | -0.380 | 0.001 |
| | MLP | -0.383 | 0.002 |
| | PosMLP | -0.382 | 0.003 |
| | S4D | -0.389 | 0.005 |
| | TCN | -0.376 | 0.004 |
| ConcentrationEasy | DNC | -0.182 | 0.004 |
| | Elman | -0.098 | 0.012 |
| | FART | -0.185 | 0.002 |
| | FWP | -0.188 | 0.001 |
| | Fr.Stack | -0.146 | 0.001 |
| | GRU | -0.039 | 0.005 |
| | IndRNN | 0.142 | 0.004 |
| | LMU | -0.057 | 0.009 |
| | LSTM | -0.080 | 0.012 |
| | MLP | -0.050 | 0.015 |
| | PosMLP | -0.048 | 0.010 |
| | S4D | -0.190 | 0.004 |
| | **TCN** | **0.253** | **0.004** |
| ConcentrationMedium | DNC | -0.182 | 0.001 |
| | Elman | -0.186 | 0.003 |
| | FART | -0.186 | 0.002 |
| | FWP | -0.185 | 0.002 |
| | Fr.Stack | -0.185 | 0.002 |
| | GRU | -0.189 | 0.003 |
| | IndRNN | -0.256 | 0.012 |
| | LMU | -0.176 | 0.005 |
| | LSTM | -0.185 | 0.001 |
| | MLP | -0.178 | 0.004 |
| | PosMLP | -0.175 | 0.002 |
| | S4D | -0.186 | 0.002 |
| | **TCN** | **-0.157** | **0.005** |
| ConcentrationHard | **DNC** | **-0.830** | **0.002** |

Continued on next page

| Env. | Model | $\mu$ | $\sigma$ |
|---|---|---|---|
| | **Elman** | **-0.830** | **0.001** |
| | **FART** | **-0.830** | **0.001** |
| | **FWP** | **-0.831** | **0.001** |
| | **Fr.Stack** | **-0.829** | **0.001** |
| | **GRU** | **-0.829** | **0.003** |
| | IndRNN | -0.849 | 0.005 |
| | **LMU** | **-0.828** | **0.002** |
| | **LSTM** | **-0.829** | **0.001** |
| | MLP | -0.833 | 0.001 |
| | **PosMLP** | **-0.831** | **0.001** |
| | **S4D** | **-0.830** | **0.002** |
| | **TCN** | **-0.830** | **0.003** |
| CountRecallEasy | DNC | -0.913 | 0.017 |
| | Elman | 0.016 | 0.091 |
| | FART | -0.300 | 0.145 |
| | FWP | -0.399 | 0.047 |
| | Fr.Stack | -0.365 | 0.022 |
| | GRU | 0.177 | 0.005 |
| | IndRNN | -0.042 | 0.050 |
| | LMU | -0.214 | 0.058 |
| | **LSTM** | **0.509** | **0.062** |
| | MLP | -0.847 | 0.011 |
| | PosMLP | -0.409 | 0.011 |
| | S4D | -0.911 | 0.005 |
| | TCN | -0.385 | 0.010 |
| CountRecallMedium | DNC | -0.907 | 0.023 |
| | Elman | -0.540 | 0.001 |
| | FART | -0.541 | 0.003 |
| | FWP | -0.541 | 0.016 |
| | Fr.Stack | -0.529 | 0.002 |
| | GRU | -0.528 | 0.001 |
| | IndRNN | -0.535 | 0.014 |
| | LMU | -0.555 | 0.009 |
| | LSTM | -0.538 | 0.005 |
| | MLP | -0.907 | 0.001 |
| | **PosMLP** | **-0.519** | **0.003** |
| | S4D | -0.920 | 0.000 |
| | **TCN** | **-0.524** | **0.003** |
| CountRecallHard | DNC | -0.815 | 0.042 |
| | Elman | -0.541 | 0.033 |
| | FART | -0.501 | 0.008 |
| | FWP | -0.477 | 0.006 |
| | Fr.Stack | -0.475 | 0.008 |
| | GRU | -0.475 | 0.006 |
| | IndRNN | -0.481 | 0.004 |
| | LMU | -0.522 | 0.036 |
| | LSTM | -0.478 | 0.006 |
| | MLP | -0.867 | 0.002 |
| | **PosMLP** | **-0.470** | **0.003** |
| | S4D | -0.858 | 0.013 |
| | **TCN** | **-0.470** | **0.002** |
| HigherLowerEasy | DNC | 0.505 | 0.002 |
| | Elman | 0.520 | 0.002 |
| | FART | 0.522 | 0.002 |

Continued on next page

|  | | $\mu$ | $\sigma$ |
| Env. | Model | | |
|---|---|---|---|
| | FWP | 0.520 | 0.002 |
| | Fr.Stack | 0.504 | 0.001 |
| | **GRU** | **0.529** | **0.002** |
| | **IndRNN** | **0.528** | **0.000** |
| | LMU | 0.494 | 0.003 |
| | **LSTM** | **0.526** | **0.003** |
| | MLP | 0.505 | 0.001 |
| | PosMLP | 0.505 | 0.001 |
| | S4D | 0.479 | 0.018 |
| | TCN | 0.503 | 0.000 |
| HigherLowerMedium | DNC | 0.501 | 0.003 |
| | Elman | 0.503 | 0.001 |
| | **FART** | **0.511** | **0.001** |
| | FWP | 0.504 | 0.006 |
| | Fr.Stack | 0.503 | 0.000 |
| | **GRU** | **0.511** | **0.002** |
| | **IndRNN** | **0.513** | **0.001** |
| | LMU | 0.466 | 0.004 |
| | LSTM | 0.504 | 0.003 |
| | MLP | 0.504 | 0.001 |
| | PosMLP | 0.505 | 0.002 |
| | S4D | 0.420 | 0.057 |
| | TCN | 0.503 | 0.001 |
| HigherLowerHard | DNC | 0.498 | 0.005 |
| | Elman | 0.501 | 0.001 |
| | **FART** | **0.507** | **0.001** |
| | FWP | 0.499 | 0.002 |
| | Fr.Stack | 0.499 | 0.002 |
| | **GRU** | **0.506** | **0.001** |
| | **IndRNN** | **0.509** | **0.001** |
| | LMU | 0.453 | 0.005 |
| | LSTM | 0.502 | 0.001 |
| | **MLP** | **0.504** | **0.002** |
| | PosMLP | 0.502 | 0.001 |
| | S4D | 0.387 | 0.037 |
| | TCN | 0.501 | 0.001 |
| LabyrinthEscapeEasy | **DNC** | **0.958** | **0.002** |
| | **Elman** | **0.956** | **0.002** |
| | FART | -0.043 | 0.266 |
| | FWP | -0.078 | 0.062 |
| | Fr.Stack | 0.521 | 0.055 |
| | **GRU** | **0.959** | **0.001** |
| | IndRNN | -0.218 | 0.136 |
| | LMU | 0.814 | 0.076 |
| | LSTM | 0.920 | 0.024 |
| | **MLP** | **0.961** | **0.000** |
| | PosMLP | 0.544 | 0.023 |
| | S4D | -0.305 | 0.033 |
| | TCN | 0.773 | 0.054 |
| LabyrinthEscapeMedium | DNC | -0.414 | 0.493 |
| | Elman | -0.122 | 0.388 |
| | FART | -0.602 | 0.057 |
| | FWP | -0.604 | 0.015 |
| | Fr.Stack | -0.384 | 0.271 |

Continued on next page

| Env. | Model | $\mu$ | $\sigma$ |
|---|---|---|---|
| | **GRU** | **0.230** | **0.642** |
| | IndRNN | -0.735 | 0.135 |
| | LMU | -0.525 | 0.081 |
| | LSTM | -0.513 | 0.123 |
| | MLP | -0.093 | 0.857 |
| | PosMLP | -0.413 | 0.049 |
| | S4D | -0.719 | 0.064 |
| | TCN | -0.494 | 0.046 |
| LabyrinthEscapeHard | DNC | -0.827 | 0.085 |
| | Elman | -0.780 | 0.039 |
| | FART | -0.828 | 0.018 |
| | FWP | -0.848 | 0.016 |
| | Fr.Stack | -0.733 | 0.040 |
| | GRU | -0.839 | 0.032 |
| | IndRNN | -0.907 | 0.030 |
| | LMU | -0.762 | 0.033 |
| | LSTM | -0.789 | 0.049 |
| | **MLP** | **-0.245** | **0.948** |
| | PosMLP | -0.806 | 0.078 |
| | S4D | -0.897 | 0.034 |
| | TCN | -0.787 | 0.011 |
| LabyrinthExploreEasy | DNC | 0.956 | 0.006 |
| | **Elman** | **0.964** | **0.001** |
| | FART | 0.424 | 0.465 |
| | FWP | -0.152 | 0.033 |
| | Fr.Stack | 0.957 | 0.001 |
| | GRU | 0.960 | 0.001 |
| | IndRNN | 0.961 | 0.001 |
| | LMU | 0.812 | 0.048 |
| | LSTM | 0.587 | 0.076 |
| | **MLP** | **0.968** | **0.000** |
| | **PosMLP** | **0.964** | **0.001** |
| | S4D | -0.265 | 0.018 |
| | TCN | 0.962 | 0.000 |
| LabyrinthExploreMedium | DNC | 0.905 | 0.005 |
| | Elman | 0.873 | 0.052 |
| | FART | -0.197 | 0.130 |
| | FWP | -0.464 | 0.041 |
| | Fr.Stack | 0.847 | 0.063 |
| | GRU | 0.893 | 0.008 |
| | IndRNN | 0.440 | 0.350 |
| | LMU | 0.423 | 0.185 |
| | LSTM | 0.025 | 0.009 |
| | **MLP** | **0.924** | **0.001** |
| | PosMLP | 0.516 | 0.191 |
| | S4D | -0.580 | 0.035 |
| | TCN | 0.903 | 0.003 |
| LabyrinthExploreHard | DNC | 0.720 | 0.212 |
| | Elman | 0.612 | 0.286 |
| | FART | -0.407 | 0.032 |
| | FWP | -0.611 | 0.044 |
| | Fr.Stack | 0.437 | 0.287 |
| | GRU | 0.796 | 0.003 |
| | IndRNN | 0.315 | 0.412 |

Continued on next page

| Env. | Model | $\mu$ | $\sigma$ |
|---|---|---|---|
| | LMU | -0.068 | 0.048 |
| | LSTM | -0.206 | 0.030 |
| | **MLP** | **0.858** | **0.002** |
| | PosMLP | -0.018 | 0.016 |
| | S4D | -0.721 | 0.008 |
| | TCN | 0.559 | 0.203 |
| MineSweeperEasy | DNC | 0.293 | 0.102 |
| | Elman | 0.640 | 0.008 |
| | FART | 0.474 | 0.030 |
| | FWP | 0.520 | 0.025 |
| | Fr.Stack | 0.516 | 0.028 |
| | **GRU** | **0.693** | **0.009** |
| | IndRNN | 0.626 | 0.003 |
| | **LMU** | **0.689** | **0.005** |
| | LSTM | 0.686 | 0.022 |
| | MLP | 0.251 | 0.019 |
| | PosMLP | 0.403 | 0.023 |
| | S4D | -0.071 | 0.024 |
| | TCN | 0.582 | 0.012 |
| MineSweeperMedium | DNC | -0.385 | 0.016 |
| | Elman | -0.009 | 0.034 |
| | FART | -0.176 | 0.020 |
| | FWP | -0.151 | 0.041 |
| | Fr.Stack | -0.177 | 0.029 |
| | GRU | -0.006 | 0.013 |
| | IndRNN | -0.024 | 0.023 |
| | LMU | -0.108 | 0.008 |
| | LSTM | -0.110 | 0.012 |
| | MLP | -0.158 | 0.006 |
| | PosMLP | -0.117 | 0.042 |
| | S4D | -0.365 | 0.019 |
| | **TCN** | **0.018** | **0.030** |
| MineSweeperHard | DNC | -0.429 | 0.002 |
| | Elman | -0.230 | 0.021 |
| | FART | -0.390 | 0.009 |
| | FWP | -0.338 | 0.036 |
| | Fr.Stack | -0.338 | 0.028 |
| | GRU | -0.206 | 0.027 |
| | IndRNN | -0.247 | 0.006 |
| | LMU | -0.294 | 0.021 |
| | LSTM | -0.303 | 0.012 |
| | MLP | -0.289 | 0.011 |
| | PosMLP | -0.278 | 0.001 |
| | S4D | -0.430 | 0.002 |
| | **TCN** | **-0.191** | **0.018** |
| MultiarmedBanditEasy | DNC | 0.302 | 0.106 |
| | **Elman** | **0.631** | **0.014** |
| | FART | 0.453 | 0.042 |
| | FWP | 0.556 | 0.045 |
| | Fr.Stack | 0.476 | 0.052 |
| | GRU | 0.619 | 0.007 |
| | IndRNN | 0.625 | 0.014 |
| | LMU | 0.332 | 0.141 |
| | LSTM | 0.527 | 0.006 |

Continued on next page

| Env. | Model | $\mu$ | $\sigma$ |
|---|---|---|---|
|  | MLP | 0.377 | 0.055 |
|  | PosMLP | 0.342 | 0.068 |
|  | S4D | 0.095 | 0.008 |
|  | TCN | 0.586 | 0.004 |
| MultiarmedBanditMedium | DNC | 0.260 | 0.066 |
|  | Elman | 0.450 | 0.043 |
|  | FART | 0.278 | 0.109 |
|  | FWP | 0.388 | 0.087 |
|  | Fr.Stack | 0.357 | 0.074 |
|  | GRU | 0.538 | 0.036 |
|  | IndRNN | 0.591 | 0.026 |
|  | LMU | 0.055 | 0.018 |
|  | LSTM | 0.476 | 0.019 |
|  | MLP | 0.350 | 0.041 |
|  | PosMLP | 0.299 | 0.055 |
|  | S4D | 0.136 | 0.006 |
|  | **TCN** | **0.598** | **0.022** |
| MultiarmedBanditHard | DNC | 0.278 | 0.075 |
|  | Elman | 0.297 | 0.103 |
|  | FART | 0.208 | 0.102 |
|  | FWP | 0.234 | 0.048 |
|  | Fr.Stack | 0.264 | 0.096 |
|  | GRU | 0.516 | 0.083 |
|  | IndRNN | 0.567 | 0.033 |
|  | LMU | 0.033 | 0.002 |
|  | LSTM | 0.419 | 0.057 |
|  | MLP | 0.391 | 0.040 |
|  | PosMLP | 0.177 | 0.028 |
|  | S4D | 0.118 | 0.024 |
|  | **TCN** | **0.574** | **0.049** |
| NoisyStatelessCartPoleEasy | DNC | 0.615 | 0.094 |
|  | **Elman** | **0.991** | **0.000** |
|  | FART | 0.983 | 0.001 |
|  | FWP | 0.966 | 0.010 |
|  | Fr.Stack | 0.856 | 0.003 |
|  | **GRU** | **0.995** | **0.000** |
|  | **IndRNN** | **0.994** | **0.001** |
|  | LMU | 0.921 | 0.020 |
|  | LSTM | 0.987 | 0.006 |
|  | MLP | 0.515 | 0.003 |
|  | PosMLP | 0.537 | 0.006 |
|  | S4D | 0.259 | 0.021 |
|  | TCN | 0.871 | 0.002 |
| NoisyStatelessCartPoleMedium | DNC | 0.312 | 0.025 |
|  | Elman | 0.640 | 0.014 |
|  | FART | 0.590 | 0.006 |
|  | FWP | 0.547 | 0.017 |
|  | Fr.Stack | 0.449 | 0.004 |
|  | GRU | 0.642 | 0.012 |
|  | **IndRNN** | **0.659** | **0.006** |
|  | LMU | 0.519 | 0.012 |
|  | LSTM | 0.638 | 0.007 |
|  | MLP | 0.302 | 0.002 |
|  | PosMLP | 0.353 | 0.004 |

| Env. | Model | $\mu$ | $\sigma$ |
|---|---|---|---|
| | S4D | 0.211 | 0.025 |
| | TCN | 0.462 | 0.007 |
| NoisyStatelessCartPoleHard | DNC | 0.233 | 0.009 |
| | Elman | 0.386 | 0.009 |
| | FART | 0.366 | 0.002 |
| | FWP | 0.354 | 0.001 |
| | Fr.Stack | 0.326 | 0.006 |
| | GRU | 0.390 | 0.007 |
| | **IndRNN** | **0.404** | **0.005** |
| | LMU | 0.352 | 0.019 |
| | LSTM | 0.393 | 0.002 |
| | MLP | 0.229 | 0.002 |
| | PosMLP | 0.288 | 0.000 |
| | S4D | 0.207 | 0.007 |
| | TCN | 0.330 | 0.004 |
| NoisyStatelessPendulumEasy | DNC | 0.456 | 0.040 |
| | Elman | 0.818 | 0.056 |
| | FART | 0.610 | 0.005 |
| | FWP | 0.532 | 0.028 |
| | Fr.Stack | 0.832 | 0.015 |
| | **GRU** | **0.894** | **0.002** |
| | IndRNN | 0.654 | 0.188 |
| | LMU | 0.654 | 0.055 |
| | LSTM | 0.786 | 0.031 |
| | MLP | 0.387 | 0.010 |
| | PosMLP | 0.467 | 0.011 |
| | S4D | 0.291 | 0.022 |
| | TCN | 0.811 | 0.015 |
| NoisyStatelessPendulumMedium | DNC | 0.435 | 0.020 |
| | Elman | 0.622 | 0.031 |
| | FART | 0.570 | 0.001 |
| | FWP | 0.521 | 0.006 |
| | Fr.Stack | 0.621 | 0.010 |
| | **GRU** | **0.717** | **0.004** |
| | IndRNN | 0.526 | 0.061 |
| | LMU | 0.570 | 0.012 |
| | LSTM | 0.653 | 0.024 |
| | MLP | 0.369 | 0.002 |
| | PosMLP | 0.446 | 0.011 |
| | S4D | 0.286 | 0.011 |
| | TCN | 0.633 | 0.013 |
| NoisyStatelessPendulumHard | DNC | 0.440 | 0.017 |
| | Elman | 0.614 | 0.017 |
| | FART | 0.553 | 0.007 |
| | FWP | 0.498 | 0.008 |
| | Fr.Stack | 0.561 | 0.008 |
| | **GRU** | **0.657** | **0.002** |
| | IndRNN | 0.521 | 0.109 |
| | LMU | 0.563 | 0.014 |
| | LSTM | 0.617 | 0.010 |
| | MLP | 0.351 | 0.012 |
| | PosMLP | 0.433 | 0.009 |
| | S4D | 0.289 | 0.011 |
| | TCN | 0.573 | 0.009 |

Continued on next page

| Env. | Model | $\mu$ | $\sigma$ |
|---|---|---|---|
| RepeatFirstEasy | DNC | 0.716 | 0.207 |
| | **Elman** | **1.000** | **0.000** |
| | **FART** | **1.000** | **0.000** |
| | FWP | 0.813 | 0.224 |
| | **Fr.Stack** | **0.997** | **0.005** |
| | **GRU** | **1.000** | **0.000** |
| | **IndRNN** | **1.000** | **0.000** |
| | **LMU** | **1.000** | **0.000** |
| | **LSTM** | **1.000** | **0.000** |
| | MLP | 0.489 | 0.391 |
| | PosMLP | 0.736 | 0.209 |
| | S4D | -0.194 | 0.098 |
| | **TCN** | **1.000** | **0.000** |
| RepeatFirstMedium | DNC | -0.349 | 0.041 |
| | Elman | -0.468 | 0.014 |
| | FART | 0.962 | 0.048 |
| | FWP | 0.830 | 0.237 |
| | Fr.Stack | -0.467 | 0.007 |
| | **GRU** | **1.000** | **0.000** |
| | **IndRNN** | **0.998** | **0.003** |
| | LMU | 0.641 | 0.457 |
| | LSTM | 0.926 | 0.068 |
| | MLP | -0.389 | 0.057 |
| | PosMLP | -0.472 | 0.007 |
| | S4D | -0.241 | 0.100 |
| | TCN | -0.449 | 0.033 |
| RepeatFirstHard | DNC | -0.334 | 0.039 |
| | Elman | -0.464 | 0.002 |
| | FART | 0.867 | 0.142 |
| | FWP | 0.915 | 0.045 |
| | Fr.Stack | -0.448 | 0.016 |
| | GRU | 0.940 | 0.012 |
| | **IndRNN** | **0.969** | **0.026** |
| | LMU | -0.406 | 0.060 |
| | LSTM | 0.275 | 0.677 |
| | MLP | -0.355 | 0.111 |
| | PosMLP | -0.455 | 0.009 |
| | S4D | -0.178 | 0.146 |
| | TCN | -0.457 | 0.010 |
| RepeatPreviousEasy | DNC | -0.223 | 0.075 |
| | **Elman** | **1.000** | **0.000** |
| | FART | 0.060 | 0.040 |
| | FWP | 0.200 | 0.052 |
| | **Fr.Stack** | **1.000** | **0.000** |
| | **GRU** | **1.000** | **0.000** |
| | IndRNN | 0.957 | 0.012 |
| | **LMU** | **1.000** | **0.000** |
| | **LSTM** | **1.000** | **0.000** |
| | MLP | -0.320 | 0.007 |
| | PosMLP | -0.336 | 0.014 |
| | S4D | -0.473 | 0.003 |
| | **TCN** | **1.000** | **0.000** |
| RepeatPreviousMedium | DNC | -0.490 | 0.001 |
| | Elman | -0.394 | 0.025 |

Continued on next page

| Env. | Model | $\mu$ | $\sigma$ |
|---|---|---|---|
| | FART | -0.468 | 0.011 |
| | FWP | -0.345 | 0.033 |
| | Fr.Stack | -0.484 | 0.003 |
| | GRU | -0.315 | 0.017 |
| | IndRNN | -0.304 | 0.014 |
| | **LMU** | **0.789** | **0.288** |
| | LSTM | -0.284 | 0.024 |
| | MLP | -0.486 | 0.001 |
| | PosMLP | -0.485 | 0.001 |
| | S4D | -0.490 | 0.002 |
| | TCN | -0.478 | 0.004 |
| RepeatPreviousHard | DNC | -0.490 | 0.002 |
| | Elman | -0.481 | 0.003 |
| | FART | -0.485 | 0.003 |
| | FWP | -0.443 | 0.017 |
| | Fr.Stack | -0.485 | 0.001 |
| | GRU | -0.428 | 0.002 |
| | IndRNN | -0.384 | 0.013 |
| | **LMU** | **0.191** | **0.041** |
| | LSTM | -0.397 | 0.008 |
| | MLP | -0.486 | 0.002 |
| | PosMLP | -0.486 | 0.004 |
| | S4D | -0.491 | 0.001 |
| | TCN | -0.486 | 0.001 |
| StatelessCartPoleEasy | DNC | 0.960 | 0.027 |
| | **Elman** | **1.000** | **0.000** |
| | **FART** | **1.000** | **0.000** |
| | **FWP** | **1.000** | **0.000** |
| | **Fr.Stack** | **1.000** | **0.000** |
| | **GRU** | **1.000** | **0.000** |
| | **IndRNN** | **1.000** | **0.000** |
| | **LMU** | **0.996** | **0.001** |
| | **LSTM** | **1.000** | **0.000** |
| | MLP | 0.722 | 0.001 |
| | PosMLP | 0.967 | 0.006 |
| | S4D | 0.514 | 0.076 |
| | **TCN** | **1.000** | **0.000** |
| StatelessCartPoleMedium | DNC | 0.956 | 0.061 |
| | **Elman** | **1.000** | **0.000** |
| | **FART** | **1.000** | **0.000** |
| | FWP | 0.989 | 0.007 |
| | **Fr.Stack** | **1.000** | **0.000** |
| | **GRU** | **1.000** | **0.000** |
| | **IndRNN** | **1.000** | **0.000** |
| | **LMU** | **0.995** | **0.001** |
| | **LSTM** | **1.000** | **0.000** |
| | MLP | 0.398 | 0.006 |
| | PosMLP | 0.593 | 0.049 |
| | S4D | 0.205 | 0.011 |
| | **TCN** | **1.000** | **0.000** |
| StatelessCartPoleHard | DNC | 0.778 | 0.330 |
| | **Elman** | **1.000** | **0.000** |
| | **FART** | **0.996** | **0.000** |
| | FWP | 0.900 | 0.092 |

Continued on next page

| Env. | Model | $\mu$ | $\sigma$ |
|---|---|---|---|
| | Fr.Stack | 0.989 | 0.017 |
| | **GRU** | **1.000** | **0.000** |
| | **IndRNN** | **1.000** | **0.000** |
| | LMU | 0.987 | 0.007 |
| | **LSTM** | **1.000** | **0.000** |
| | MLP | 0.265 | 0.002 |
| | PosMLP | 0.443 | 0.018 |
| | S4D | 0.127 | 0.026 |
| | **TCN** | **0.998** | **0.003** |
| StatelessPendulumEasy | DNC | 0.427 | 0.022 |
| | Elman | 0.903 | 0.007 |
| | FART | 0.652 | 0.010 |
| | FWP | 0.556 | 0.007 |
| | Fr.Stack | 0.906 | 0.005 |
| | **GRU** | **0.913** | **0.002** |
| | IndRNN | 0.798 | 0.151 |
| | LMU | 0.819 | 0.107 |
| | LSTM | 0.878 | 0.048 |
| | MLP | 0.448 | 0.005 |
| | PosMLP | 0.486 | 0.011 |
| | S4D | 0.290 | 0.007 |
| | TCN | 0.906 | 0.003 |
| StatelessPendulumMedium | DNC | 0.436 | 0.012 |
| | **Elman** | **0.880** | **0.004** |
| | FART | 0.660 | 0.013 |
| | FWP | 0.579 | 0.033 |
| | **Fr.Stack** | **0.881** | **0.002** |
| | **GRU** | **0.884** | **0.002** |
| | IndRNN | 0.719 | 0.216 |
| | LMU | 0.858 | 0.003 |
| | LSTM | 0.875 | 0.005 |
| | MLP | 0.455 | 0.025 |
| | PosMLP | 0.509 | 0.011 |
| | S4D | 0.282 | 0.003 |
| | TCN | 0.874 | 0.002 |
| StatelessPendulumHard | DNC | 0.420 | 0.033 |
| | Elman | 0.819 | 0.005 |
| | FART | 0.698 | 0.077 |
| | FWP | 0.663 | 0.051 |
| | **Fr.Stack** | **0.824** | **0.002** |
| | **GRU** | **0.828** | **0.001** |
| | IndRNN | 0.804 | 0.023 |
| | LMU | 0.806 | 0.006 |
| | LSTM | 0.819 | 0.006 |
| | MLP | 0.477 | 0.030 |
| | PosMLP | 0.543 | 0.020 |
| | S4D | 0.303 | 0.014 |
| | TCN | 0.822 | 0.005 |

