# OpenReview forum: "POPGym: Benchmarking Partially Observable Reinforcement Learning"
_ICLR.cc/2023/Conference — ICLR 2023 poster_

### Official Review · Reviewer_HVVV · 2022-10-18

**Confidence:** 4
**Correctness:** 4
**Technical Novelty And Significance:** 1
**Empirical Novelty And Significance:** 2
**Recommendation:** 8

**Clarity, Quality, Novelty And Reproducibility:**

The clarity of the work is good.

The quality is fair; however, as articulated in the weaknesses section, there is substantial room for improvement.

The originality is good in the respect that there has not been much work organizing POMDP benchmarks for deep RL.

The reproducibility is high in the sense that code is available and appears to be well organized but less high in the sense that outlier-sensitive results were reported over a small number of runs.

**Strength And Weaknesses:**

### Strengths

- I agree with the submission's characterization that there is a lack of suitable partially observable testbeds.
- Because the submission compares existing algorithms, rather than introducing new ones, the experiments are less likely to suffer from experimenter bias.
- From a brief perusal, the code appears easy to read and well organized.

### Weaknesses

- While (again) I agree that there is a dearth of good POMDP benchmarks, I am not sure the submission does a great job addressing the problem:
    - Six of the 14 games are small modifications of fully observable settings
    - Four are described as "diagnostic", meaning useful as "memory introspection and debugging tools"
    - Two are relatively simple card games,
    - Two are navigation based, which, as the submission points out, are one of the few types of POMDPs for which there are suitable existing benchmarks.

    What I am getting at is that the partial observability in POPGym largely doesn't feel *strategically interesting*.
- The other gripe I have is with algorithm benchmarking:
    - First, given that the submission is a benchmarking paper, it seems disappointing that it implements but does not benchmark differentiable neural computers. The submission states that this is because computation cost, despite also indicating that POPGym environments induce relatively quick convergence.
    - Second, the metrics that were reported are not up to date with modern deep RL benchmarking -- see *Deep RL at the Edge of the Statistical Precipice* (NeurIPS 2021).
        - The most obvious criticism here is that each algorithm is over only three seeds. It is well documented that three seeds may be insufficient to accurately reflect performance. Given that it is a benchmark paper, I'd hope for at least 5 or 10.
        - Additionally, it would be good to include the metrics endorsed by the statistical precipice paper, such as interquartile mean, optimality gap, performance profiles, and stratified bootstrap confidence intervals.

**Summary Of The Paper:**

The submission introduces a library containing 14 POMDPs, each with three levels of difficulty, and 13 memory model baselines. The submission benchmarks 12 of these memory baselines.

**Summary Of The Review:**

While the idea motivating the submission (providing good POMDP benchmarks for deep RL) is one that is very valuable to the community, I think the submission falls short of its potential both in terms of the proposed environments and the benchmarking of existing algorithms.

Still, I think the submission ought to be accepted, as the value it provides to the community is strictly positive.

I hope the authors will consider addressing the weaknesses listed above in future revisions.

---

> ### Author Response · Authors · 2022-11-11
> **Response to HVVV**
>
> >While (again) I agree that there is a dearth of good POMDP benchmarks, I am not sure the submission does a great job addressing the problem:
> >* Six of the 14 games are small modifications of fully observable settings
> >* Four are described as "diagnostic", meaning useful as "memory introspection and debugging tools"
> >* Two are relatively simple card games,
> >* Two are navigation based, which, as the submission points out, are one of the few types of POMDPs for which there are suitable existing benchmarks.
> >What I am getting at is that the partial observability in POPGym largely doesn't feel _strategically interesting_.”
>
> We have edited the description of diagnostic tasks to be more clear. The diagnostic tests are still quite difficult environments, they just tend to focus on quantifiable aspects of memory, e.g. being able to reconstruct a sequence of observations, determining if memory can forget information after it becomes useless, etc.
>
> We intend to add more environments as time goes on. After submission, we implemented, tested, and benchmarked the Count Recall environment. We posted a call for new POMDP suggestions on our public repo.
>
> We are not sure we understand what you mean by __”strategically interesting”__, could you clarify? The review period is not enough to implement, test, and benchmark new environments, but if you have any POMDP suggestions, we would be happy to start implementing them.
>
> >It seems disappointing that it implements but does not benchmark differentiable neural computers. The submission states that this is because computation cost, despite also indicating that POPGym environments induce relatively quick convergence.
>
> We started new differentiable neural computer (DNC) experiments the weekend the reviews were released, but please note that in Figure 2 these are orders of magnitude slower than other memory models, regardless of how fast our environments are. The DNC runs are roughly 50% done at the time of writing (Thursday Nov. 10th). We will do our best to update the paper with these runs before the review closes, otherwise we will add them to the final release, if accepted.
>
> >It is well documented that three seeds may be insufficient to accurately reflect performance. Given that it is a benchmark paper, I'd hope for at least 5 or 10.
>
> We agree that more experiments would be ideal, however we have already run over 1700 separate trials and our compute is fully tied up with the requested DNC experiments. We specifically chose conservative hyperparameters for our experiments, so the standard deviations and confidence intervals reported in the appendix are quite small in most cases. Furthermore, performance over a number of separate environments (now 15) should provide good signal, even if a single environment result might contain some uncertainty. We are unfortunately limited in what we can do during the review period.
>
> >Additionally, it would be good to include the metrics endorsed by the paper, such as interquartile mean, optimality gap, performance profiles, and stratified bootstrap confidence intervals.”
>
> We now provide the interquartile range, mean, and median in the summary plot (Fig. 3) and removed the summary table that only provided means. In the appendix, we report the mean, standard deviation (table), and the 95% bootstrapped confidence interval (bar chart) for all environments and all models. We also report the reward curves with a 95% bootstrapped confidence interval in the appendix.

---

> > ### Comment · Reviewer_HVVV · 2022-11-11
> > **Thanks for your reply!**
> >
> > > We intend to add more environments as time goes on. After submission, we implemented, tested, and benchmarked the Count Recall environment. We posted a call for new POMDP suggestions on our public repo.
> >
> > That's great to hear.
> >
> > > We are not sure we understand what you mean by ”strategically interesting”, could you clarify?
> >
> > One example of a POMDP that I would consider strategically interesting is computing a best response to a distribution of partners in Hanabi. This setting is "strategically interesting" in the sense that, to do well, it is necessary to (at least implicitly) do inference about the likelihood of partners and private hands, given the observed behavior, in order to optimize the forward-looking expected return. This inference involves complexity beyond simply remembering an old observation.
> >
> > > The review period is not enough to implement, test, and benchmark new environments, but if you have any POMDP suggestions, we would be happy to start implementing them.
> >
> > Agreed. I also don't think the Hanabi-based POMDP I described is a suitable fit for the submission's benchmark, given how expensive it would be.
> >
> > > We started new differentiable neural computer (DNC) experiments the weekend the reviews were released, but please note that in Figure 2 these are orders of magnitude slower than other memory models, regardless of how fast our environments are. The DNC runs are roughly 50% done at the time of writing (Thursday Nov. 10th). We will do our best to update the paper with these runs before the review closes, otherwise we will add them to the final release, if accepted.
> >
> > Awesome!
> >
> > > We agree that more experiments would be ideal, however we have already run over 1700 separate trials and our compute is fully tied up with the requested DNC experiments. We specifically chose conservative hyperparameters for our experiments, so the standard deviations and confidence intervals reported in the appendix are quite small in most cases. Furthermore, performance over a number of separate environments (now 15) should provide good signal, even if a single environment result might contain some uncertainty. We are unfortunately limited in what we can do during the review period.
> >
> > Makes sense. I hope the authors will consider running a larger number of seeds at some point, even if not during the review period.
> >
> > > We now provide the interquartile range, mean, and median in the summary plot (Fig. 3) and removed the summary table that only provided means. In the appendix, we report the mean, standard deviation (table), and the 95% bootstrapped confidence interval (bar chart) for all environments and all models. We also report the reward curves with a 95% bootstrapped confidence interval in the appendix.
> >
> > Thanks for doing this. I hope the authors will consider including additional metrics suggested by the statistical precipice paper, even if not during the review period.
> >
> > ---
> >
> > As stated in my original review, I think the value of the submission is strictly positive. I do not see any good reason for gatekeeping it from publication at ICLR.

---

### Official Review · Reviewer_X3ta · 2022-10-25

**Confidence:** 4
**Correctness:** 4
**Technical Novelty And Significance:** 3
**Empirical Novelty And Significance:** 2
**Recommendation:** 8

**Clarity, Quality, Novelty And Reproducibility:**

The writing is quite clear and in good quality in my opinion. Below are some minor comments.

1. This it would be worthwhile to add that current benchmarks test the high dimensional observations.
2. Figure 3. Can you normalize the reward such that 1 would indicate the optimal value? if I understand correctly it seems that the normalisation might make it seems the performance is high although it is extremely sub optimal (this can happen due to the normalization).
3. End of section 1.1. should put period "." at the end of each line.
4. page 3, "wall following behavior", what does that mean?
5. page 5 "Like Repeat Fist" -> "Like Repeat First"
6. Page 5. Reference to additional bandit book: https://tor-lattimore.com/downloads/book/book.pdf.

**Strength And Weaknesses:**

Pros:
Partially observed system are a crucial challenge to tackle for applying RL in real-world applications. I tend to agree with the authors claim that currently there are not enough benchmarks to test different RL algorithms for the partially observable settings. I hope that this work will encourage further research into the partially observable setting, and lead to the design of new and practical algorithms for this domain.

Cons:
Although the authors introduced several new environments there are several aspects of partially observed systems that are not being investigated. First, the problems (in my opinion) are quite basic and still seem quite far from practical applications. Second, the theoretical RL community recently studied several POMDP models that would be interesting to explore from practical perspective (in my opinion):
1. Weakly revealing POMDPs (see https://arxiv.org/pdf/2204.08967.pdf).
2. MDPs with latent context (see https://arxiv.org/abs/2102.04939).
3. \gamma observable POMDP (see https://arxiv.org/abs/2206.03446)



**Summary Of The Paper:**

The authors introduced a set of new environments to test RL algorithms on partially observable environments. The authors introduced a diverse set of environments in which different aspects of the hardness of partially observed system was tested. Further, the authors conduced quite extensive study and tested the performance of previously suggested algorithms for the partially observed domain.


**Summary Of The Review:**

Overall, I think this paper has a clear and nice contribution: it introduced new benchmarks for study partially observed RL problems, This, in my opinion, is very much worthwhile and will hopefully encourage the community to study these type of problems in more depth. Further, I hope that the authors/additional contributers will expand the set of task in the future.

---

> ### Author Response · Authors · 2022-11-11
> **Response to X3TA**
>
> > First, the problems (in my opinion) are quite basic and still seem quite far from practical applications
>
> We agree that our environments are far from practical application. Environments mentioned in the related work, such as DMLab, are closer to real-world application than our proposed environments. However, there is a trade off – more realistic and practical environments have higher latency and require training larger models. The point of POPGym is to provide __fast__ and __simple__ environments, similar to those in the original OpenAI Gym library. POPGym should serve as a proving ground: new memory models can be quickly validated and tuned on POPGym before training on a more expensive and complex target environment. As we note in the related work, most POPGym environments can be solved in a few hours while more complex simulators take weeks.
>
> >The theoretical RL community recently studied several POMDP models that would be interesting to explore from practical perspective (in my opinion):
> >1. Weakly revealing POMDPs (see https://arxiv.org/pdf/2204.08967.pdf).
> >2. MDPs with latent context (see https://arxiv.org/abs/2102.04939).
> >3. \gamma observable POMDP (see https://arxiv.org/abs/2206.03446)
>
> We appreciate your suggestions. We now clarify that our environments are overcomplete POMDPs in the POPGym Environments section. The rebuttal time is too short for us to come up with, implement, test, and benchmark new environments. Nonetheless, we have published an issue to our public repo asking for suggestions and we will make a good-faith effort to implement any suggestions. Promises are worth little on the internet, but while the paper was under review, we implemented, tested, and benchmarked a new environment (Count Recall). Hopefully, this provides evidence that we intend to maintain and incorporate new POMDPs into our library as time goes on.
>
> Moving on, the work from Ni et al. provide a number of latent MDPs, but they call it “Meta RL” so we didn’t make the connection at first. We now accurately describe their environments as “Latent MDPs”, pointing potentially interested users to them in the Related Work section.
>
> With regards to gamma POMDPs: Assigning value to an observation is a very powerful tool that we can use to make strong claims about POMDPs. Unfortunately, it seems that this measure is conditioned on the belief state, so it measures the observability of an environment with respect to a model. It seems quite difficult to measure the observational value when our belief states are uninterpretable, which is the case for all of our benchmarked models.
>
> > Minor comments
>
> Thank you for the minor comments. We have integrated all of them into the paper.

---

> > ### Comment · Reviewer_X3ta · 2022-11-18
> > **Thanks for the response!**
> >
> > I hope that the library will be maintained and improved in the future to support the study of partially observed problems in RL.

---

### Official Review · Reviewer_eVQA · 2022-10-25

**Confidence:** 4
**Correctness:** 3
**Technical Novelty And Significance:** 2
**Empirical Novelty And Significance:** 2
**Recommendation:** 3

**Clarity, Quality, Novelty And Reproducibility:**

The clarity is high, the quality is decent, novelty is on a borderline level. The reproducibility is good.

**Strength And Weaknesses:**

Strength
+ A very good and in-depth summary of the current RL benchmarks, especially on the partially observable environments. After analysis of the shortcomings of the current benchmarks, they motivate the work;
+ The author conducted an extensive set of experiments with 14 partially observable environments and with 13 memory model baselines;
+ Considerations have been made for wide adoption of the proposed benchmark, including low compute cost for a wide community, compatible APIs with the popular training library, etc;

Weaknesses
- I would expect the analysis part of a benchmark paper would take the largest portion of the paper and will have a very detailed analysis of the benchmarks and experiments. However, the major parts of the paper are about the task description and baseline description;
- Though the authors have run very extensive experiments, the results do not seem dramatically different than what people already understood;
- I  found limited insights from the benchmark results;
- The authors should point out potential directions for tackling the partially observable tasks after the benchmarking.

**Summary Of The Paper:**

The paper performs benchmarks on partially observable RL environments and a bunch of network architectures on memory. The authors make sure the learning of the POPGym environments is affordable with consumer-grade GPU to ensure the experiments are feasible for researchers. They conclude that GRU is the best general-purpose memory model.

**Summary Of The Review:**

I found unsatisfactory that a benchmark paper does not provide much content on the in-depth analysis of experiment results, even though the authors conduct an extensive set of experiments.

---

> ### Author Response · Authors · 2022-11-11
> **Response to eVQA**
>
> >I would expect the analysis part of a benchmark paper would take the largest portion of the paper and will have a very detailed analysis of the benchmarks and experiments. However, the major parts of the paper are about the task description and baseline description
>
> Our original submission followed the tradition of prior literature on RL benchmarks, which focuses on the code/library and quantitative results, rather than analysis (e.g. ProcGen [1], DMLab [2], and Arcade Learning Environment [3]). That said, we have completely rewritten the discussion and conclusion sections to clarify and emphasize our findings.
>
> [1] https://proceedings.mlr.press/v119/cobbe20a/cobbe20a.pdf
>
> [2] https://arxiv.org/pdf/1612.03801.pdf
>
> [3] https://www.jair.org/index.php/jair/article/view/10819/25823
>
> >Though the authors have run very extensive experiments, the results do not seem dramatically different than what people already understood; I found limited insights from the benchmark results
>
> We have emphasized our non-intuitive findings in our revised discussion section. To summarize a few of our less-intuitive findings:
> 1. Modern POMDP benchmarks rely heavily on navigation tasks. We hypothesize and then show that MLPs routinely outperform all memory models on nearly all navigation tasks. As far as we know, nobody else has made this connection explicit in RL. This has far-reaching implications for how we measure progress on partially observable RL.
> 2. Time-dependent stateless policies (positional MLP) are much more effective than they have any right to be, nearly solving Stateless Cartpole and outperforming MLPs in many cases.
> 3. We show that supervised learning results are a poor predictor of memory in RL, providing a need for POMDP benchmarks. Despite the explosion of attention in RL, we show it exhibits relatively poor performance when used as memory.
> 4. Before our study, there were few/no RL-based studies on the IndRNN, FART, LMU, and S4D baselines. It was not clear how these models would perform against other baselines.
> 5. Elman RNNs, discarded decades ago for their vanishing/exploding gradients, work surprisingly well in RL – on-par with LSTM in fact, while being more efficient in every sense of the word.
>
> >The authors should point out potential directions for tackling the partially observable tasks after the benchmarking.
>
> This is a good point. We have revised our text so that nearly every paragraph in the discussion section now provides avenues for future research.

---

> ### Comment · Reviewer_HVVV · 2022-11-11
> **Re: Weaknesses**
>
> > I would expect the analysis part of a benchmark paper would take the largest portion of the paper and will have a very detailed analysis of the benchmarks and experiments. However, the major parts of the paper are about the task description and baseline description;
>
> This is maybe a matter of personal taste, but I didn't have any issues with the structure of the submission. The tasks and baselines are important parts of benchmarking papers.
>
> > Though the authors have run very extensive experiments, the results do not seem dramatically different than what people already understood; I found limited insights from the benchmark results;
>
> I'm not sure that I agree that this is a weakness. The implicit insinuation here is that that authors would have benefitted from fudging their results into looking more surprising.
>
> That said, I actually did find some of their results surprising. For example, I did not anticipate an MLP to perform as well as it did on tasks that clearly require memory; also, I did not necessary anticipate a GRU to outperform an LSTM to the extent to which it did.
>
> > The authors should point out potential directions for tackling the partially observable tasks after the benchmarking.
>
> I do not mind benchmark papers doing this if the experimenters feel they have gained insight into promising future directions from the process of their experiments. However, I do not think benchmarkers ought to be compelled to offer such ideas if they do not have any. Benchmarking by itself is both an important and under-pursued contribution.

---

### Author Response · Authors · 2022-11-11
**Summary Response**

We thank all the reviewers and chairs for taking the time to review our paper, and we are very positively encouraged  that others find POMDPs as meaningful and exciting as us. We are excited to hear that we provide a “very good and in-depth summary (of RL benchmarks)” and an “extensive set of experiments” (Reviewer eVQA). Reviewer X3TA writes that “partially observed systems are a crucial challenge”, that we provide, a “clear and nice contribution”, and that our benchmark “is very much worthwhile and will hopefully encourage the community to study these type of problems in more depth.” Reviewer HVVV notes that “the value [the submission] provides to the community is strictly positive.”

Reviewer eVQA has concerns that our discussion section is lacking. We address eVQA’s analysis concerns with a complete rewrite of the discussion and conclusion sections. Our analytical contributions should now be much more clear.

X3TA and HVVV share concerns about our distribution of provided POMDPs. After reading papers provided by X3TA, we explicitly categorize our available environments as overcomplete POMDPs [1,2], and now correctly point users to Ni et al. [3] which contains a library of latent MDPs. We published a public issue on our GitHub repo requesting environment suggestions for types of POMDPs we do not yet represent in our library. We will make a good-faith effort to implement any suggested environments that adhere to the efficiency principles of POPGym. We stress that our library will be updated and actively maintained for at least the next two years – while our submission underwent review we have already added, tested, and benchmarked a new environment (Count Recall) across all memory baselines.

[1] https://proceedings.neurips.cc/paper/2017/hash/6aca97005c68f1206823815f66102863-Abstract.html

[2] https://arxiv.org/abs/2006.12484

[3] https://proceedings.mlr.press/v162/ni22a.html

---

### Public Comment · ~Kazuki_Irie1 · 2023-04-15
**Depth is crucial for attention/Transformer-based memory models**

Congratulations on this very nice work!

I just wanted to point out that the following statement in the paper is misleading:

> *"Although linear attention methods like FWP and FART show significant improvements across a plethora of supervised learning tasks, they were some of the worst contenders in RL"*

As far as I understood, only single-layer variants are tested here for all models except IndRNN (*"The FART and FWP models use a single attention block"*).
It is not surprising that this setting favours RNN models: attention-only/Transformer models typically require **many more layers** to perform well as powerful memory models in general.
Even for language modelling (see, e.g., https://arxiv.org/abs/1905.04226) or supervised learning tasks, single-layer (or in general, shallow) Transformers do not perform well (typically underperform shallow LSTMs).

So I expect that, also for RL in PO environments, Transformer variants should be much deeper to perform to their full potential!
(Edit: For example, in Table 3 of this paper https://arxiv.org/abs/2106.06295, we compare 2-layer vs. 4-layer recurrent FWPs in the environments where the LSTM baseline outperforms the 2-layer variant.)

PS: removing the positional encodings also only makes sense for auto-regressive Transformers with at least two layers!

---

> ### Author Response · Authors · 2023-04-17
> **Limitations are based on the recurrent state size**
>
> Hello, thank you for the comment. With so many disparate models, it is hard to do a fair comparison. We have no doubt that the FWP with a larger recurrent size or more layers would likely do better. In Appendix A, we write:
>
> > There is no clear axis for a truly fair comparison between memory models – model throughput and parameter count vary drastically throughout the memory models. We decide to limit the amount of storage (i.e. recurrent state size) of each memory model to 256 dimensions, which is greater than the common values of 64 and 128 in literature (Ni et al., 2022). This results in lower parameter counts for models that produce the recurrent state using a tensor product (e.g. FWP, FART, S4D). We could make an exception for these models, allowing them to produce recurrent states of size 256^2 = 65336 dimensions instead of 16^2 = 256 dimensions to bring up the parameter count, but we believe this is unfair to recurrent models. In this case, recurrent models would need to forget and compress information over longer episodes, while tensor product models could store every single observation in memory without any compression or forgetting. Storing everything is unlikely to scale to real-world applications where episodes could span hours, days, or run indefinitely.
>
> Splitting 256 dimensions across 2 or more attention layers would result in very few weights (11 for each key/query/value in a 2-layer setup or 8 each in a 4-layer setup). Thus, we figured one layer would be better.
>
> We tested the model both with and without positional encodings, finding the single-layer FWP did better with encodings, so we used that configuration for the experiments.
>
> In [your paper](https://arxiv.org/pdf/2102.11174.pdf), both the original (simple summed KV) and improved models (DPFP+delta update) are called FWP. We do not implement the delta update rule and utilize ReLU in place of DPFP, so perhaps it is confusing that we call it FWP.
>
> Here is what I propose:
> 1. I will update the paper to make it clear we are not using DPFP/delta-updates, and only the "basic" FWP/linear transformer with sum normalization.
> 2. I will add a disclaimer about small recurrent states and refer to Appendix A in the main text
> 3. I will add multi-layer support for all linear transformers in POPGym, so that users may evaluate them with whatever depth/size they want
> 4. I will add additional feed-forward layers to the linear transformers to help mitigate the effects of small K/Q/V dims
> 5. I will rerun all linear transformer experiments with four attention layers and no positional encoding. If they perform better than the one layer model, I will update the results accordingly.

---

> > ### Public Comment · ~Kazuki_Irie1 · 2023-04-18
> > **Thank you very much for your reply!**
> >
> > I had not expected this detailed/elaborated reply: thank you very much!
> >
> > Your proposals 1-5 sound great to me. Regarding 1., yes, that clarification is important: in particular, the use of the delta rule typically makes a big difference, as you can see in the cited papers and their follow-ups (while DPFP is not part of our standard setting anymore).
> > Regarding 4., the architecture to follow is that of the Transformer (i.e., with 2-layer feedforward blocks, layernorms, and residual connections, where appropriate). Essentially, FWP layers should simply replace the standard self-attention layers in the Transformer.
> >
> > PS: Just for the sake of completeness, I’m also providing some more clarifications on the subtleties in my original comment.
> >
> > > *With so many disparate models, it is hard to do a fair comparison.*
> >
> > I completely understand, and I’m sorry that I had missed that paragraph in the appendix!
> > My original comment was referring to the fact that (1) the “statement/conclusion” you found (that I quoted above) is not specific to RL. Under such an “equal-state size” constraint for comparison, even for supervised learning or language modelling, Transformer-family models have little chance to outperform the RNN models. And therefore, (2) it is misleading to leave that “statement” as a general conclusion.
> >
> > > *We have no doubt that the FWP with a larger recurrent size or more layers would likely do better.*
> >
> > My comment was not at all a general/superficial criticism of the type *“why not make it bigger*?”
> > One-layer variant is conceptually/computationally very limited as a standalone memory model. I do not expect a one-layer model to perform “much better” even when its size is increased. Depth is really crucial for a Transformer-family model to *become* a powerful sequence processor.
> >
> > > *We tested the model both with and without positional encodings, finding the single-layer FWP did better with encodings, so we used that configuration for the experiments.*
> >
> > I had understood this. What I meant in the PS of my original comment is that this result (“positional encodings help one-layer FWPs”) was expectable, because, conceptually, auto-regressive Transformer models need *at least 2 layers* so that they can encode positional information on their own (through "auto-regressiveness“). One-layer models do need some extra/explicit signals for positional information.
> >
> > Thank you so much once again for your reply!

---

> > > ### Author Response · Authors · 2023-06-28
> > > **Preliminary results are in**
> > >
> > > Dear Kazuki,
> > >
> > > Apologies for the delay. We have updated the code in the repo to enable multilayer attention, as well as added additional feed-forward layers and a residual connection for our tested attention models. We ran two variants of linear transformer experiments: 1 attention layer with positional encoding and 4 attention layers without encoding.
> > >
> > > We found the linear attention models performed better with the extra feed-forward layers and residual connection. However, the 1 layer variant still outperforms the 4 layer variant given a constant recurrent state size, so we will replace the scores with the 1 layer updated scores. The mean score for FWP has increased from ~0.11 to ~0.18.
> > >
> > > We have added text making it clear we do not implement the delta update, as well as a sentence in the main text linking to the recurrent size constraints in the appendix. It will take some time to regenerate all the plots in the paper, so we will post again once this is done and the new version is uploaded.

---

> > > > ### Public Comment · ~Kazuki_Irie1 · 2023-07-27
> > > > **Thank you again**
> > > >
> > > > I'm sorry for my very late reply. I've only seen this now as OpenReview does not send any notifications to non-authors (even to the person who initially posted the public comment)...
> > > >
> > > > Thank you for all the updates. This is much, much more than I initially expected.
> > > >
> > > > Thank you very much once again for your thorough and elaborated responses.

---

### Decision · Program_Chairs · 2023-01-20

**Decision:**

Accept: poster

**Justification For Why Not Higher Score:**

The paper is clearly missing deep analysis on understanding each given result.

**Justification For Why Not Lower Score:**

With the same reason above, I would be OK if the paper is rejected -- and give one more iteration to deeply analyze and polish.

**Metareview: Summary, Strengths And Weaknesses:**

This paper analyzes multiple POMDP algorithms on numerous environments with different aspects. While their experimental and discussion sections could benefit from more extensive comparisons and deeper analysis, the paper has tested and compared various algorithms on different environments. This benchmark of POMDPs is actually quite important and timely -- many real worlds problems almost always require working with partial observation, thus understanding the current states of algorithms adds an important value to the community. I however personally found their results not super surprising, but it is also good to share with the community that many results follow our intuition or word-of-mouth. So, I give a weak accept. Should the paper be rejected, I highly recommend performing more analysis/ablation studies.

**Note From Pc:**

if the above contains the word "oral" or "spotlight" please see: "oral" presentation means -> notable-top-5% and "spotlight" means -> notable-top-25%. As stated in our emails, we are disassociating presentation type from AC recommendations